# Clonal dynamics in osteosarcoma defined by RGB marking

Stefano Gambera[1], Ander Abarrategi [1,2], Fernando González-Camacho[3], Álvaro Morales-Molina[1], Josep Roma [4], Arantzazu Alfranca [1,5] & Javier García-Castro [1]

Osteosarcoma is a type of bone tumour characterized by considerable levels of phenotypic heterogeneity, aneuploidy, and a high mutational rate. The life expectancy of osteosarcoma patients has not changed during the last three decades and thus much remains to be learned about the disease biology. Here, we employ a RGB-based single-cell tracking system to study the clonal dynamics occurring in a de novo-induced murine osteosarcoma model. We show that osteosarcoma cells present initial polyclonal dynamics, followed by clonal dominance associated with adaptation to the microenvironment. Interestingly, the dominant clones are composed of subclones with a similar tumour generation potential when they are re-implanted in mice. Moreover, individual spontaneous metastases are clonal or oligoclonal, but they have a different cellular origin than the dominant clones present in primary tumours. In summary, we present evidence that osteosarcomagenesis can follow a neutral evolution model, in which different cancer clones coexist and propagate simultaneously.

[1] Cellular Biotechnology Unit, Instituto de Salud Carlos III (ISCIII), Madrid 28029, Spain. [2] Haematopoietic Stem Cell Lab, The Francis Crick Institute, London NW1 1AT, UK. [3] Electron and Confocal Microscopy Unit, Instituto de Salud Carlos III (ISCIII), Madrid 28029, Spain. [4] Laboratory of Translational Research in Child and Adolescent Cancer, Vall d'Hebron Hospital, Barcelona 08035, Spain. [5] Immunology Department, Hospital Universitario de La Princesa, Madrid 28006, Spain. These authors contributed equally: Arantzazu Alfranca, Javier García-Castro. Correspondence and requests for materials should be addressed to J.G.-C. (email: jgcastro@isciii.es)

Osteosarcoma (OS) is the most common malignant solid tumour that affects bones. The disease presents a bimodal distribution with increased incidence during the second decade of life; OS represents more than 10% of solid cancer cases in adolescents (15–19 years old)[1]. The paediatric incidence window reflects the biology of the disease; there is a correlation between skeletal growth, height, and disease appearance. Moreover, OS usually originates in the extremities of long bones, close to the metaphyseal plate, which is also the anatomical location of bone growth[2]. Almost 75% of OS is highly malignant, and due to disease aggressiveness, it has typically extended beyond the bone into nearby musculoskeletal structures at detection[1,2]. Tumour biopsies showing mesenchymal cells producing osteoid and/or irregular woven bone are categorized as OS. The histologic finding of this incomplete osteogenic process is a requirement for tumour diagnosis even if other cell subtypes directly derived from the tumour are present. This pathological definition is used because the aetiology of OS is mostly unknown. Genetic disorders, such as Li–Fraumeni syndrome (TP53 germline mutation) and familial Retinoblastoma (RB1 germline mutation), are risk factors for osteosarcoma[3,4]. The Pediatric Cancer Genome Project (PCGP) identified frequent germline mutations of the TP53 gene in OS, similar to the 50% TP53 mutation rate of childhood cancers[5,6], and whole genome and whole exome sequencing revealed that alterations in the p53 and Rb pathways are more frequent in OS than previously thought[7,8]. Therefore, these syndromes are mainly associated with mutations of genes that participate in genome integrity maintenance and chromosomal stability. Unlike many sarcomas, which are characterized by specific chromosome translocations, OS exhibits a complex karyotype with high genomic and chromosomal instability;[9] it is also characterized by multiple rearrangements across the genome, kataegis, and chromothripsis[8,10–12].

Malignant tumours typically comprise a heterogeneous pool of cells that differ in terms of morphology, phenotype, gene expression, metabolism, immunogenicity, proliferation, and metastatic potential[13,14]. Many models have been postulated to explain the clonal dynamics that drive cancer disease and the generation of heterogeneity[14,15]. The competitive linear model of clonal cancer evolution proposed by Nowell[16] and the cancer stem cell hypothesis were the first models describing cancer evolution[17–19]. Later, other authors suggested that these two models were not mutually exclusive because cancer stem cells could be the unit of selection during cancer initiation and progression. A switch from differentiation to self-renewal, supported by the niche, can generate compartment amplification, in which cancer stem cell units can also undergo independent evolution[13,20,21]. With the advent of cancer genome studies, branched phylogenies were adopted to describe cancer evolution[22–25]. Additionally, the sequential accumulation of genetic alterations was recently questioned due to evidence indicating macroevolutionary events[14,26]. Other authors have rejected clonal dominance in favour of a big bang model of clonal diversity, in which different clonal cancer populations are generated early in tumourigenesis and coexist with neutral evolution dynamics[27,28]. In this context, the ecological interaction between tumour subclones[29–31] and the dynamics of contingency, convergence, and parallel evolution are implicated in tumour growth[14]. In the current view of the cancer ecosystem, non-genetic determinants also contribute to tumour growth. The interaction between tumour cells and the microenvironment, differentiation programs, factors such as hypoxia, and especially the immune system represent crucial players in cancer development[14,21]. Another largely unexplored field of clonal cancer dynamics concerns metastatic development. From the seed and soil hypothesis and the preferential diffusion pathway of some tumours, the modern definition of a pre-metastatic niche highlights the importance of the microenvironment in metastatic cell tropism to seed-specific organs[32]. Some studies have shown a monoclonal pattern of metastatic seeding, but others have reported a polyclonal signature for this process[33].

A model that exhaustively describes cancer growth is extremely important because this knowledge has many practical implications in the clinic. Especially in the field of personalized medicine, the clonal homogeneity of a primary tumour and heterogeneity of metastatic cells represent relevant variables for designing a therapeutic strategy. A single tumour biopsy may be insufficient to provide representative information of the total genetic and molecular variability present in the primary tumour. Additionally, the implication of heterogeneity in the management of patients presenting with metastatic disease represents a significant challenge. The general approach, driven by the assumption of close similarity between a primary tumour and metastases, has been to analyse the primary tumour and avoid more invasive biopsies at metastatic locations. This approach restricted the estimation of how many different clones can constitute a tumour or metastasize to an organ. Moreover, metastatic disease is a time-dependent process; nevertheless, little is known about its timing, the changes in the clonal composition over time, and the degree of independent evolution between primary tumour and metastases.

To study the events driving osteosarcomagenesis, here, we focus on the clonal dynamics that occur during the formation, development, and progression of a murine model of in vitro transformed mesenchymal progenitor cells (MPCs). We previously reported the transformation of MPCs by deleting the Tp53 and Rb genes. These MPCs, if inoculated in the proper orthotopic or ectopic ceramic-based osteoinductive microenvironment, efficiently recapitulate OS formation[34]. In this study, we used a single-cell tracking technique to study the in vivo OS clonal dynamics during tumour formation and progression. Based on lentiviral transduction with vectors coding for three different fluorescent proteins (Cerulean, Venus, and Cherry) as a marking approach (Lenti LeGO-RGB vectors), we developed a protocol in which each individual OS cell displays a different colour of the rainbow spectrum. These cells were used to interrogate the clonal evolution-related questions in in vivo orthotopic, ectopic, and metastatic tumourigenesis studies. In our studies we show that osteosarcomagenesis can follow a neutral evolution model; different clones can coexist and propagate over time and only some of them become locally dominant invading the adjacent microenvironment. Metastatic disease also presents signs of polyclonality, where metastatic clones can be distinct from the dominant clones present in the primary tumour. In summary, our study offers an overview of the clonal dynamics in OS development.

## Results

**Efficient and stable RGB marking of murine MPCs.** To generate RGB (red–green–blue) multi-coloured cells, we used three lentiviral vectors Cherry (red), Venus (green), and Cerulean (blue), which express different fluorescent proteins. The RGB marking of murine $p53^{-/-}Rb^{-/-}$ bone marrow-derived MPCs (BM-MPCs) was achieved by transducing cells with multiplicity of infection (MOI) corresponding to equimolar transduction efficiency per vector of 50%. Correct cell line transduction was validated based on fluorescent colour saturation and the variability of colour mixing. As verified by confocal microscopy, an optimal colour spectrum was obtained using an MOI of 0.75 per vector, whereas excessively high MOIs resulted in poor colour mixing (Supplementary Figure 1a). These multi-coloured cells,

designated RAINBONE cells in this text, showed a wide range of colours, with each colour representing a different clone (Fig. 1a). To optimize flow cytometry analysis, monoclonal cell lines were obtained by the limiting dilution of RAINBONE cells, and an oligoclonal mix was further generated by mixing six of these monoclonal lines. As shown in Supplementary Figure 1b, each monoclonal cell line presented a narrow peak of fluorescence, whereas the oligoclonal mix was composed of a combination of discrete peaks (Supplementary Figure 1c). In contrast, polyclonal RAINBONE cells displayed a broad fluorescent distribution generated by the integration of signals from a multitude of clonal populations (Supplementary Figure 1d). Three-dimensional (3D) visualization, which was accomplished by plotting the three fluorescent variables in a Cartesian plot ($x$,$y$,$z$), or 3D plot, increased clonal discrimination. Thus, monoclonal or oligoclonal cell lines could be easily identified, whereas heterogeneous RAINBONE cells covered the three axes and their possible fluorescent colour combinations.

Clonal heterogeneity and the stability of fluorescent markers of RAINBONE cells were studied and quantified during 50 days of in vitro culture by flow cytometry (Supplementary Figure 2; Fig. 1b, c). Visual stochastic network embedding (viSNE) (Supplementary Figure 3a) and spanning-tree progression analysis of density-normalized events (SPADE) (Supplementary Movie 1) were also applied to study clonal heterogeneity over time. Our results showed that the multicolour spectrum of RAINBONE cells was stable and that clonal heterogeneity was maintained during in vitro culture.

**Tumour heterogeneity in osteosarcomas**. It was previously shown that murine $p53^{-/-}Rb^{-/-}$ BM-MPCs can generate OS if implanted ectopically in the proper osteoinductive microenvironment[34]. Therefore, we tested the nature of subclonal interaction in tumour growth. RAINBONE cells were further transduced with an ff-Luciferase lentiviral vector, and unmarked $p53^{-/-}Rb^{-/-}$ BM-MPCs were also transduced to calculate an optimal MOI and obtain >80% transduction efficiency (Supplementary Figure 4). Limiting dilution clones obtained with RAINBONE cells and a clonal mix of increasing clonal composition were embedded in ceramic scaffolds and implanted subcutaneously in vivo in NOD.Cg-Prkdcscid-Il2rgtm1Wjl/SzJ (NSG) mice; luciferase activity was used to directly quantify tumourigenicity and tumour growth kinetics (Supplementary Figure 5a). Seven out of 7 randomly selected monoclonal cell lines were tumourigenic (100% incidence), further revealing a tendency to grow faster if compared to OS generated by a clonal mixture of either 5 or 10 different limiting dilution clones or the pool of RAINBONE cells. Overall, OS tumour growth was slower at increasing clonal complexity and each clone shows a heterogeneous growth kinetic if implanted alone. Furthermore, around 40% of RAINBONE cells formed colonies at in vitro cell transformation assays (Supplementary Figure 5b).

In summary, this experiment confirmed the competitive nature of subclonal populations and further indicated that the Ad-Cre deletion of $Tp53$ and $Rb$ is a strong transforming event for murine MPCs which are composed of a pool of heterogeneous transformed cells.

Given their competitive nature, the clonal composition of RAINBONE-generated OS was studied in vivo for short (25 days) and long (50 days) periods. An experimental workflow schematic is shown in Fig. 2a. After 25 days, RAINBONE cells developed tumour masses with highly vascularized areas surrounding the ceramics. Histologically, tumours developed heterogeneously, with both rich bone matrix deposition areas and fibroblastic regions (Fig. 2b). At this stage of development, tumour cells

showed no clonal dominance by flow cytometry (Fig. 2c) or confocal studies (Fig. 2d, e and Supplementary Figure 6a). Instead, tumours were characterized by large areas of multi-coloured cells suggesting a polyclonal composition. viSNE (Supplementary Figure 3b) and SPADE analysis (Supplementary Figure 3c) also confirmed clonal heterogeneity. Genome insertion site analysis by linear amplification-mediated polymerase chain reaction (LAM-PCR) indicated strong amplification of different long terminal repeat (LTR)-genome junctions (Fig. 2f and Supplementary Figure 7), further supporting a polyclonal tumour composition. Spectral karyotyping (SKY) analysis identified a high level of genomic heterogeneity among cancer cells, which presented a tetraploid karyotype with high levels of aneuploidy, some large deletions, and non-clonal translocations (Supplementary Figure 8 and Supplementary Table 1).

In summary, our data demonstrate that at an early stage, OS can be composed of coexistent, heterogeneous, competing cancer populations. However, this competition is not associated with a strong clonally selective event.

**Clonal evolution in osteosarcoma progression**. To further explore the dynamics of OS progression, we extended tumour development and clonal studies. In contrast to the previous results, RAINBONE tumours showed changes in their clonal growth after a longer period (50 days). The central tumour mass maintained a polyclonal composition, but the tumour periphery showed abundant expansion of a few clones (Fig. 3a and Supplementary Figure 6b). A heterogeneous clonal composition was confirmed by LAM-PCR (Fig. 3c and Supplementary Figure 7), and flow cytometry detected the enrichment of some discrete populations. This result was also confirmed by SPADE and viSNE analysis (Fig. 3e and Supplementary Figure 3b–c). Histologically, extracompartmental areas showed increased cellularity with scant bone matrix deposition, whereas an osteogenic phenotype was maintained in the ceramic region (Supplementary Figure 6b). Of note, clones having the ability to grow outside the ceramic region were different in each animal tumour (Fig. 3e). Despite these areas being apparently monoclonal, a few clones could also be identified, supporting the heterogeneous nature of these regions (Fig. 3d). Explanted tumour cells from these peripheral regions were sorted by fluorescence-activated cell sorting (FACS) based on their fluorescence fingerprint (Fig. 3f). As expected, these clones showed a few discrete insertion sites in a pattern similar to the control monoclonal populations obtained by in vitro limiting dilution (Fig. 3g; Supplementary Figure 7 and Supplementary Table 2). SKY analysis identified high genomic heterogeneity among cancer cells even when they were of monoclonal origin (Supplementary Figure 8 and Supplementary Table 1). Additionally, for these clones, the clone-to-clone relationship presented a competitive nature; 4 out of 4 monoclonal tumours in secondary transplantation exhibit faster growth compared to an oligoclonal mix composed of the previous tumour (Supplementary Figure 5c). However, subclonal competition did not lead to clonal extinction; all clones remained present, as indicated by flow cytometry analysis (Supplementary Figure 5d–left). Three out of 4 oligoclonal tumours showed the dominance of the G11 clone (turquoise arrow), whereas one tumour was mostly formed by the R7 population (pink arrow), which surprisingly showed faster growth when implanted alone (Supplementary Figure 5c). Furthermore, in vitro growth assays indicated that in experimental conditions with no space competition and an equal nutrient supply, the R9 clones grew faster (Supplementary Figure 5d–right); nevertheless, this clone was infrequently identified in the in vivo assay (Supplementary Figure 5d–left).

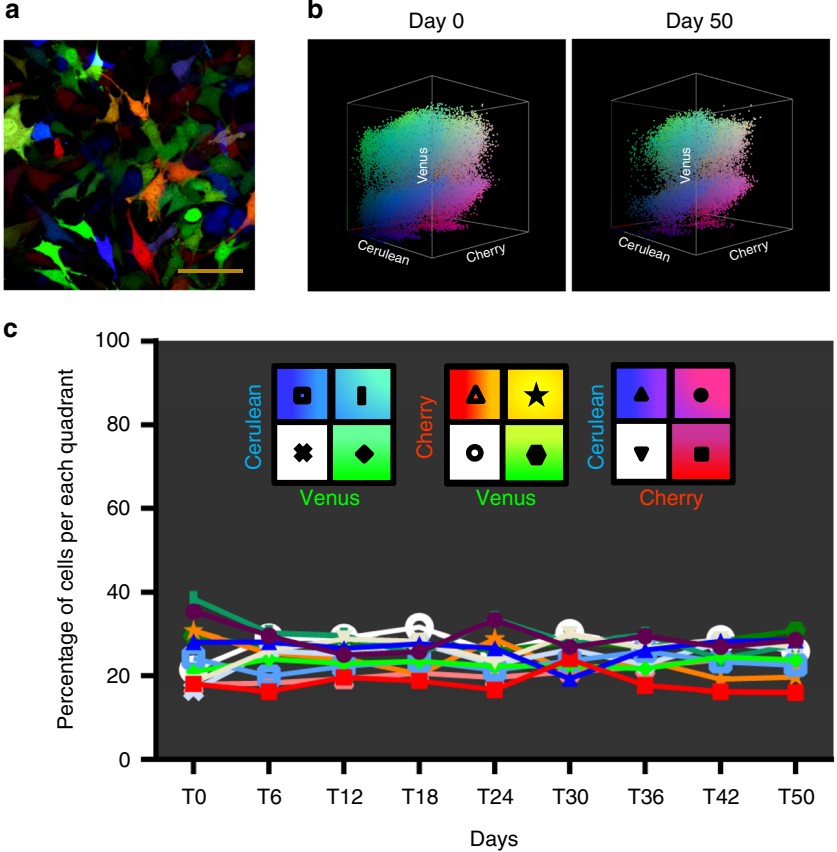

**Fig. 1** RGB marking of RAINBONE cells is stable during in vitro culture. **a** Representative confocal microscopy image of the optimal colour mixture. Orange bar = 100 µm. **b** Representative 3D flow cytometry analysis of RAINBONE cells at day 0 (left) and day 50 (right) of in vitro culture. **c** Quantification of the fluorescent distribution during 50 days of in vitro culture

In summary, these experiments demonstrate that clonal evolution is associated with the ability to grow in a new microenvironment at a late stage of disease. Furthermore, this phenotype is acquired by different dominant clones that present karyotypic heterogeneity and different in vitro or in vivo growth rates. In this case, we confirmed a regional and selective clonal dominance while a highly polyclonal area remains present.

**Neutral and selective dynamics of orthotopic OS development.** Tumour formation at the orthotopic site by our experimental $p53^{-/-}Rb^{-/-}$ BM-MPCs was previously demonstrated to efficiently recapitulate the main features of OS, including metastatic disease[34]. Therefore, RAINBONE cells were also inoculated in the proximal tibia of immunodeficient mice, and tumour development was evaluated by bioluminescence and X-ray imaging. Mice showed radiographic characteristics compatible with intramedullar bone deposition and cortical bone osteolysis (Fig. 4a, d). At 50 days after RAINBONE cells were implanted, mice were killed due to limb function loss. Confocal microscopy indicated great colour heterogeneity in the tumour. Cells of different clonal origins were found in the medullar space infiltrating the compact bone, growing at the endosteum, and even as sparse single cells at perivascular locations (Fig. 4b). Increased pseudo-trabecular bone formation was promoted by cells of different clonal origins and also evidenced by colourful osteoblastic rimming (Fig. 4c). In some mice, tumours produced a strong periosteal reaction with structures resembling Codman triangles and presented a soft tissue mass development over the bone surface (Fig. 4d). These areas were composed of few dominant clones with increased invasiveness of the adjacent musculoskeletal tissues (Fig. 4e).

Tumours frequently destroyed the metaphyseal plate (Fig. 4f) and developed outside of the medullar cavity in large monoclonal globular-shaped soft tissue masses, which also presented low-frequency infiltrating clones of different clonal origins (Fig. 4e, g).

Altogether, the results suggest that clonal heterogeneity is a common growth dynamic in our OS models and that there are signs of clonal evolution in the late phases, a characteristic that is primarily associated with the ability to grow in a new microenvironment. Furthermore, we show that orthotopic and osteoinductive ectopic models do not differ substantially in terms of clonal evolution; each develops some clones that are able to expand extracompartmentally in late-stage disease.

**Metastatic disease is driven by polyclonal seeding of lungs.** Bioluminescent and X-ray imaging were employed in the orthotopic model as diagnostic high-sensitivity techniques. Bioluminescence revealed the formation of multiple metastatic nodules in the lungs, and this was further confirmed by histological analysis (Fig. 5a and Supplementary Figure 6c). Metastatic dissemination appeared heterogeneous, with different clonal development dynamics. Metastases were usually heterogeneous, presenting different sizes and clonal origins (Fig. 5b). In total, 146 metastases were quantified for monoclonality or oligoclonality and measured (Fig. 5c). Monoclonal seeding was more frequent; however, excluding small micrometastases (<200 µm), which could represent a dormant state, the nodules showing significant growth were both monoclonal and oligoclonal (Fig. 5d). Furthermore, metastatic clones did not correspond to the dominant clones present in the primary OS tumour (Fig. 5e and Supplementary Figure 9a).

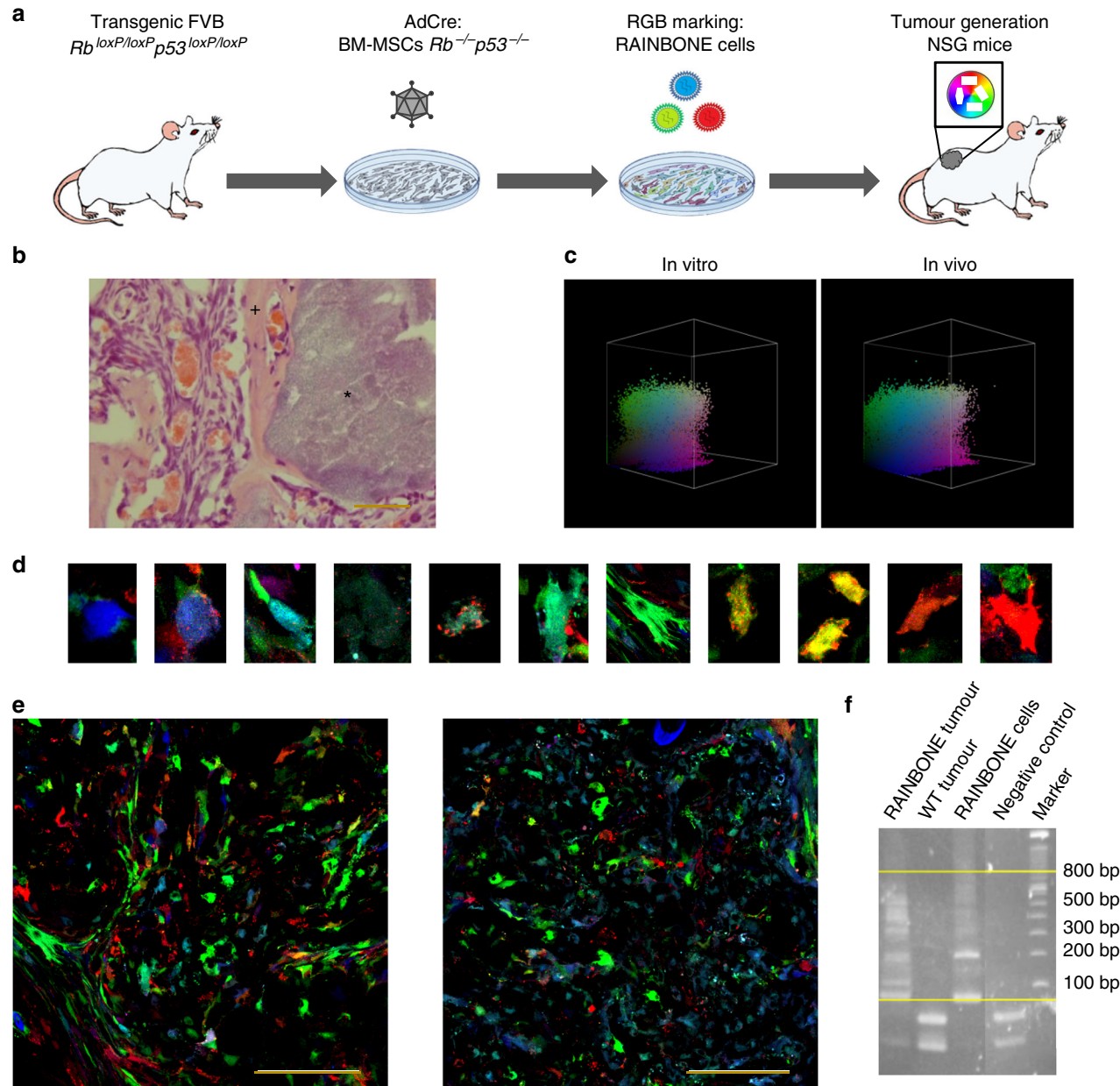

**Fig. 2** RAINBONE cells generate polyclonal tumours in ectopic implants. **a** Experimental scheme employed for ectopic OS development ($n = 7$). **b** Haematoxylin/eosin staining (HE) staining of tumours generated by ectopically implanted cells. Spindled cells produced abundant bone matrix ($+$) near HA/TCP ceramics ($*$). **c** Representative 3D flow cytometry analysis of in vitro RAINBONE cells (left) and explanted tumour cells at day 25 (right) ($n = 3$). **d** Representative confocal images showing the colour spectra of different in vivo tumour clones. **e** Representative confocal macroscopic view of polyclonal RAINBONE tumours indicating the contribution of different clones to tumour generation; orange bars $= 100\,\mu m$. **f** LAM-PCR analysis showing the amplification of different provirus integration sites in RAINBONE tumours. Picture represents the same gel; vertical lines were added where non-informative lanes were excised. Horizontal yellow lines demarcate the standard size range of the PCR products to exclude amplification artefacts. Negative control $=$ water, WT tumour $=$ tumour generated by unmarked cells

Assuming that heterogeneous nodules could originate from the lung homing of a cluster of cells or by secondary clones homing into a pre-existing nodule, we decided to test the ability of in vivo tested metastatic clones (Supplementary Figure 9b) to seed pre-existing metastases. We induced RAINBONE tumours, and bioluminescence was used to follow tumour growth and effective lung engraftment (Fig. 5f). Individual metastatic clones were inoculated intravenously (i.v.) when there was evidence of spontaneous metastatic disease (50 days). After 10 days of i.v. inoculation, lungs were dissected and processed for confocal studies. Only four out of 75 (5.3%) lung metastatic nodules analysed presented the characteristic fluorescent fingerprint of the i.v. inoculated clones, indicating the low homing ability of metastatic clones into pre-existing metastatic lung nodules (Fig. 5g).

Our data indicate that metastatic disease is highly heterogeneous and that different clones develop in the lungs with signs of polyclonal seeding. Moreover, the aggressiveness of dominant clones at the primary site does not correlate with increased metastatic clonal frequency.

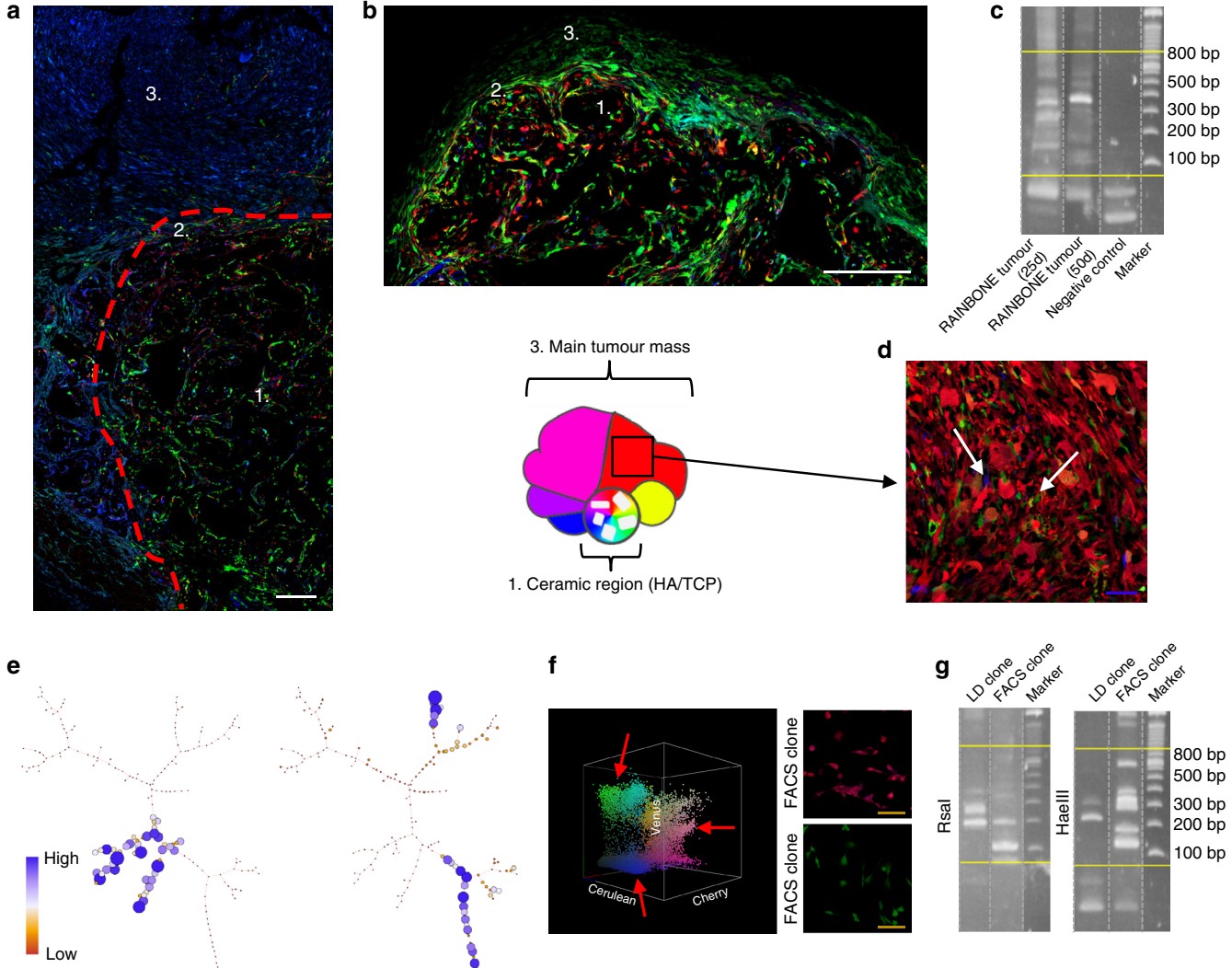

**Fig. 3** RAINBONE tumours showed dominant clones in the long term. **a** Representative image of tumours 50 days after RAINBONE inoculation ($n = 14$). 1, Primary implantation site (HA/TCP ceramic region) showing polyclonal evolution; 2, transition zone; 3, large monoclonal areas generated outside of the ceramic region (extracompartmental region). **b** For comparison, representative image of tumours at 25 days, with tumour growth starting in the extracompartmental region. **c** LAM-PCR analysis showing the rich amplification of different provirus integration sites in both short- and long-term RAINBONE tumours. Negative control = water. **d** Representative detailed image of tumour heterogeneity present in the large clonal area. Low-frequency clones are indicated by white arrows. **e** Representative SPADE tree clonal composition of RAINBONE tumours at 50 days. Note that in this case, colours represent the population frequency. **f** Representative 3D flow cytometric RAINBONE tumour analysis at 50 days ($n = 4$). Arrows indicate the most frequent clones that were subsequently FACS sorted (right). **g** LAM-PCR analysis showing the monoclonal origin of FACS sorted clones ($n = 3$). White bar = 200 μm; orange bar = 100 μm; blue bar = 50 μm

**Dominant clones are composed of homogeneous subclones**. As described above, RAINBONE tumours induced subcutaneously in osteogenic implants present spatially dominant clone development at late-stage disease (50 days). We wanted to study the grade of heterogeneity of these dominant clones, including the hypothetical presence of cancer stem cells responsible for sustaining tumour repopulation. Therefore, we designed an experiment (Fig. 6a) in which dominant clones were sorted by FACS and monoclonal populations were established (Fig. 6b, c); three populations were further decoloured using adenoviral vectors expressing Cre recombinase. Fluorescent marker loss was assessed by flow cytometry (Fig. 6d), and we did not observe resistance to Ad-Cre recombination; three out of 3 clones showed almost pure fluorescent marker loss (Supplementary Figure 10 and Supplementary Table 3). Decoloured cells underwent a second round of RGB colouring using lentiviral Gene Ontology (LeGO)-RGB lentiviral vectors, generating RAINBONE-2 cells (Fig. 6e, f),

which were implanted in NSG mice. In these secondary tumours, we observed a strong reduction in bone matrix content and faster tumour development (15 days) compared to primary RAIN-BONE tumours. Confocal study and flow cytometry with secondary tumours showed a polyclonal contribution to tumour development (Fig. 6g, h). Then, RAINBONE-2 explanted cells underwent further tertiary transplantation in NSG mice. Again, these tertiary tumours showed the same polyclonal heterogeneity (Fig. 6i, j). The viSNE and SPADE analysis also confirmed the heterogeneous subclonal composition (Supplementary Figure 11). Our results indicate that dominant clones are formed by a homogeneous equilibrium of subclones with similar tumour regeneration potential.

## Discussion

Single-cell studies and massive genome sequencing techniques have allowed the tracking of tumour development[14]. The results

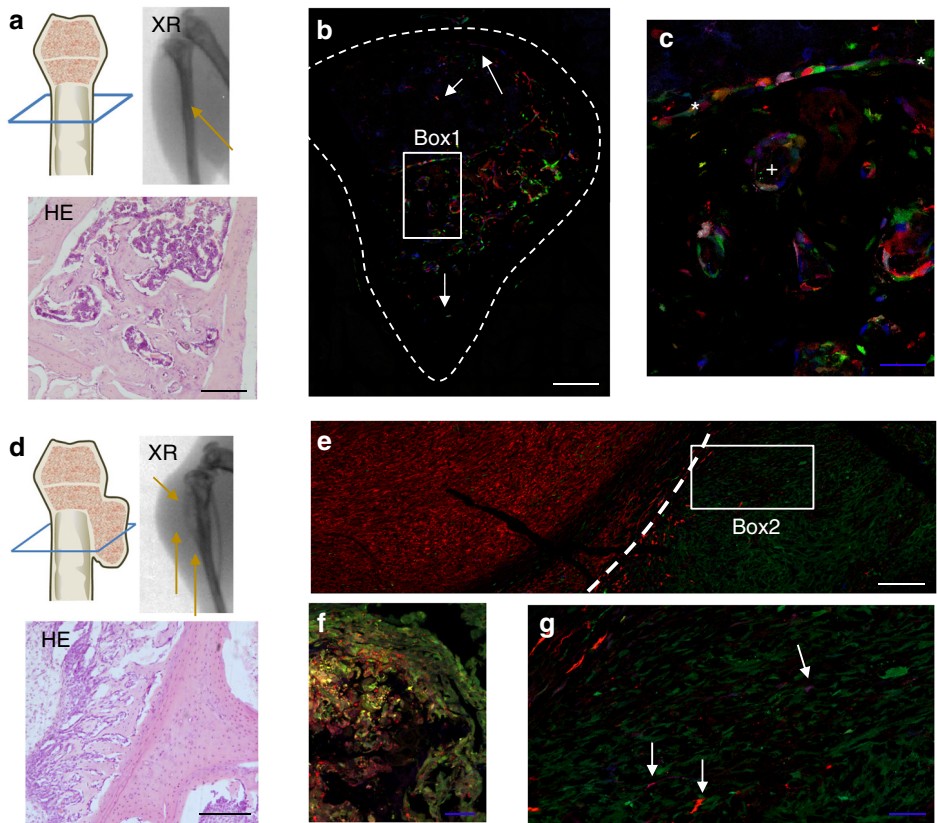

**Fig. 4** Orthotopically implanted RAINBONE cells generate polyclonal tumours. **a** Tumour developing in the medullar space (orange arrow). Blue section indicates the visual plane for haematoxylin/eosin staining (HE) and confocal images. XR X-ray imaging. **b** Representative image of RAINBONE cells located in the endosteal space, at perivascular locations, and in compact bone (white arrows). **c** Detail view of picture B (Box 1) showing osteoblastic rimming (*) and malignant trabecular bone (+). **d** Osteolytic tumour overgrowing as a soft tissue mass at 50 days presented strong periosteal reaction and a Codman triangle (orange arrows). **e** Representative image of a large bi-clonal area growing on the outer surface of bone; two different dominant clones are developing in opposition (white line indicates the border limits). **f** Detailed view of cartilage plate infiltration and destruction. **g** Picture detail (Box 2) of the clonal growth in the surrounding of musculoskeletal tissue, with low-frequency clones (white arrow) among larger monoclonal areas. White and black bars = 200 μm; blue bars = 50 μm; $n = 5$

of these studies provide better understanding of cancer as a heterogeneous disease and highlight differences in the growth patterns of specific tumour types. Here, we used a single-cell tracking technique based on fluorescent protein expression using lentiviral vectors (Lenti LeGO-RGB vectors). Due to the variety of integration sites and vector copy number, when these vectors are used in the appropriate combination, they mark each individual cell with a different colour of the rainbow spectrum that is then transmitted to derived progeny. This technology is a powerful tool for clonal cell studies in vitro and in vivo[35–37] as it represents an unbiased approach for studying tumour physiology; it does not require any preselected marker and allows the direct study of tumour clones and progeny in the spatial organization of the tissue. Lenti LeGO-RGB marking has been successfully used to clonally track in vivo metastatic mammary adenocarcinoma[38], pancreatic adenocarcinoma[39], and neuroendocrine carcinoma[35], and it has also been used in combination with mass spectrometry[39].

These findings support the value of RGB marking in tumour heterogeneity studies. However, this approach has only been tested in well-established carcinoma cell lines. Murine $p53^{-/-}Rb^{-/-}$ BM-MPCs employed for generating RAINBONE cells were not isolated from pre-existing tumours but were transformed in vitro prior to inoculation into mice. Our double-hit model allows murine cell transformation with the establishment of a heterogeneous cell population of transformed MPCs. In our conditions,

RAINBONE cells present malignant features and are efficiently transformed by $p53$ and $Rb$ loss. This result is in contrast with previous studies reporting a low tumour-initiating potential for mesenchymal lineage cells. However, in our studies we employed severe immunodeficient NSG mice and implanted cells in an orthotopic and ectopic bone-like microenvironment, thus excluding some of the harsh conditions that most likely affected other studies. Indeed, the tumour-initiating cell frequency of melanoma cells can reach 25% of the total population when using a cell matrix or less immunocompetent mice[40]. Given the high tumour-initiating potential, our model represents a powerful tool to test the cohabitation of different cancer clones and the possible dynamics of competition among them (see further discussion below). Our artificial condition represents a model with which we can test clonal evolution, the existence of selective events, or even the neutral dynamics of growth. Furthermore, these cells hypothetically have not been shaped by the tumour microenvironment and have never experienced the growth dynamics and selective pressure occurring during in vivo tumour development. All these characteristics make RAINBONE cells an interesting model of in vivo primary tumour generation, allowing us to test different hypotheses about clonal dynamics and dispersal forces occurring during osteosarcomagenesis.

Our experience with RGB marking is that it represents a very powerful technology; nevertheless, some technical difficulties were extremely challenging. We were unable to rapidly isolate a

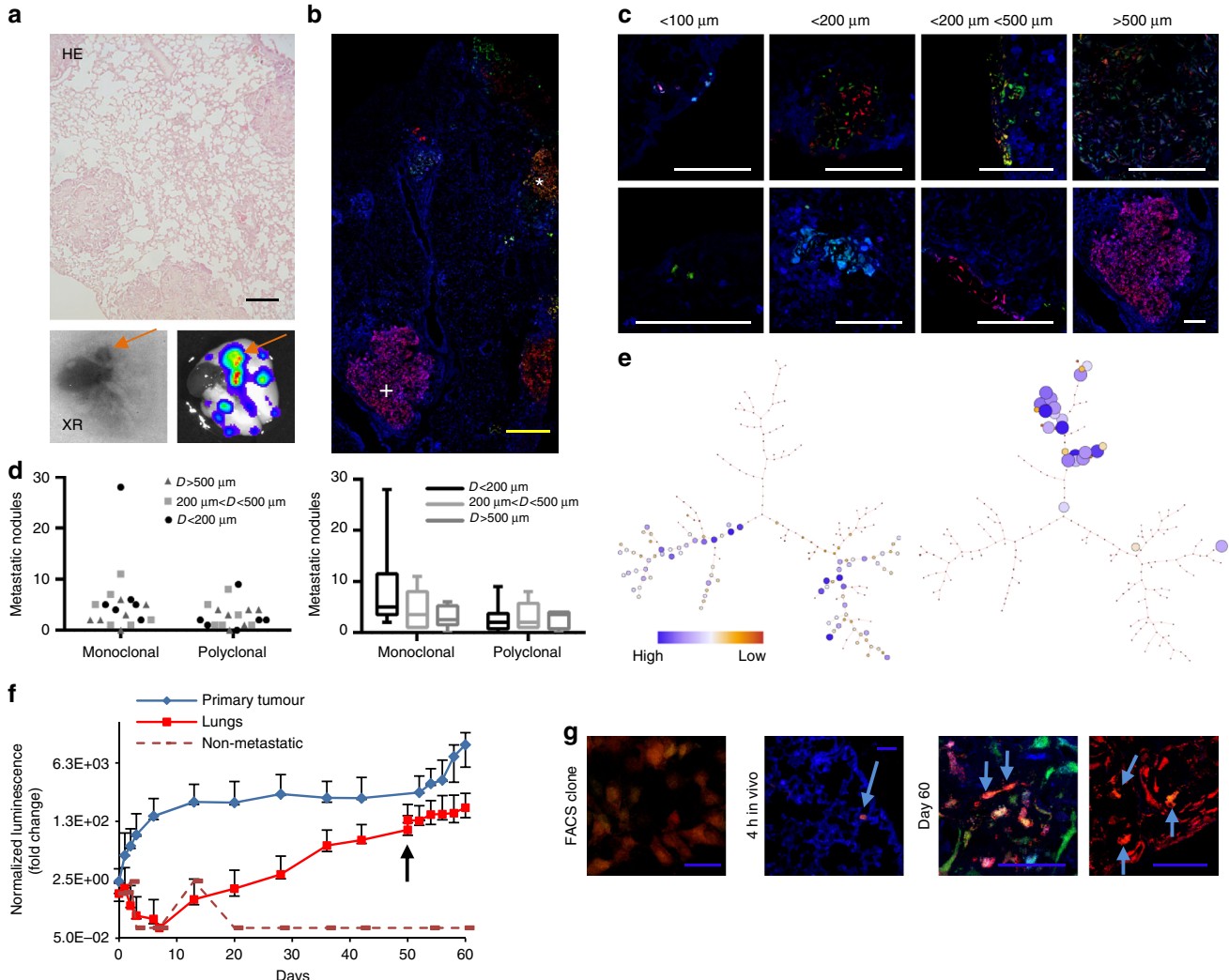

**Fig. 5** RAINBONE cells generate monoclonal and polyclonal lung metastases. **a** Orthotopic OS tumours generated lung metastasis (*n* = 5, same mice from the previous experiment) that was detected by haematoxylin/eosin staining (HE), X-ray (XR), and bioluminescent imaging. Orange arrows indicate ossified lung metastasis. **b** Representative image of RAINBONE lung metastases showing nodules of different colours and composition; monoclonal (+) and oligoclonal (*) nodules. **c** Representative images of oligoclonal and monoclonal nodules and size heterogeneity. **d** Quantification of metastatic nodules (*n* = 146) according to monoclonality and polyclonality (left graph, *P* value = 0.0497) and further categorized depending on their size (right graph, *P* values = 0.396). Statistic tests: two-tail paired *t*-test and two-way ANOVA, Bonferroni post hoc test, CI: 95%, alpha: 0.05. *P* < 0.05. In boxplot graphs centre line indicates median, bounds of box indicate 25th and 75th percentiles, and whiskers indicate minimum and maximum. **e** Representative SPADE tree clonal composition of explanted primary tumour (left) and lung cells (right). In this case, colour represents the population frequency. **f** Bioluminescent in vivo growth quantification of primary tumours (*n* = 5, new experimental group) and lung metastases (*n* = 4). One animal did not develop metastatic disease (dotted line). The black arrow indicates intravenous inoculation of monoclonal FACS sorted clones in reseeding experiments (50 days). The average radiance was normalized to the bioluminescent signal 4 h after inoculation (day 0); data are presented as the means and the error bars represent the maximum and minimum values per group. **g** From the left to the right: representative confocal image of in vitro FACS clones, cells (blue arrows) after 4 h of i.v. inoculation, and two different reseeded metastases at day 60; for those appearing in colour, a further colocalization study was performed. Yellow bar = 500 μm; white bar = 200 μm; blue bar = 50 μm

specific monoclonal population by FACS sorting RAINBONE tumours. This limitation, produced by a loss of definition in flow cytometric data, was due to the equal representation of different clonal populations with overlapping flow cytometric fingerprints. Furthermore, in some cases it was difficult to obtain a pure clonal sorting, and some very low-frequency clones of different colours appeared in culture. We tried different flow cytometers, services, and users, and we solved this problem by resorting clones after a short in vitro amplification. We rationalized that sorting multiple clonal populations at the same time can affect process efficiency and purity. In conclusion, with the employment of different microscopic and genetic techniques, we avoided the

misinterpretation of results, and the technical limitations did not greatly hinder the main objective of the study.

In evolutionary theories, competition is a long and steady principle that is continuously occurring in ecological systems, such as cancer[41,42]. From an early stage of cancer development, tumour cells compete for limited resources (nutrients and oxygen) to the point of saturation[21] and encounter a strongly selective microenvironment (pH changes, immune system, geographic barriers, and chemotherapy) that further limits their development[14]. In this context, dispersal forces could also play a key role in tumour progression;[43] populations of spreading cells could reach untapped resources, increasing distance from

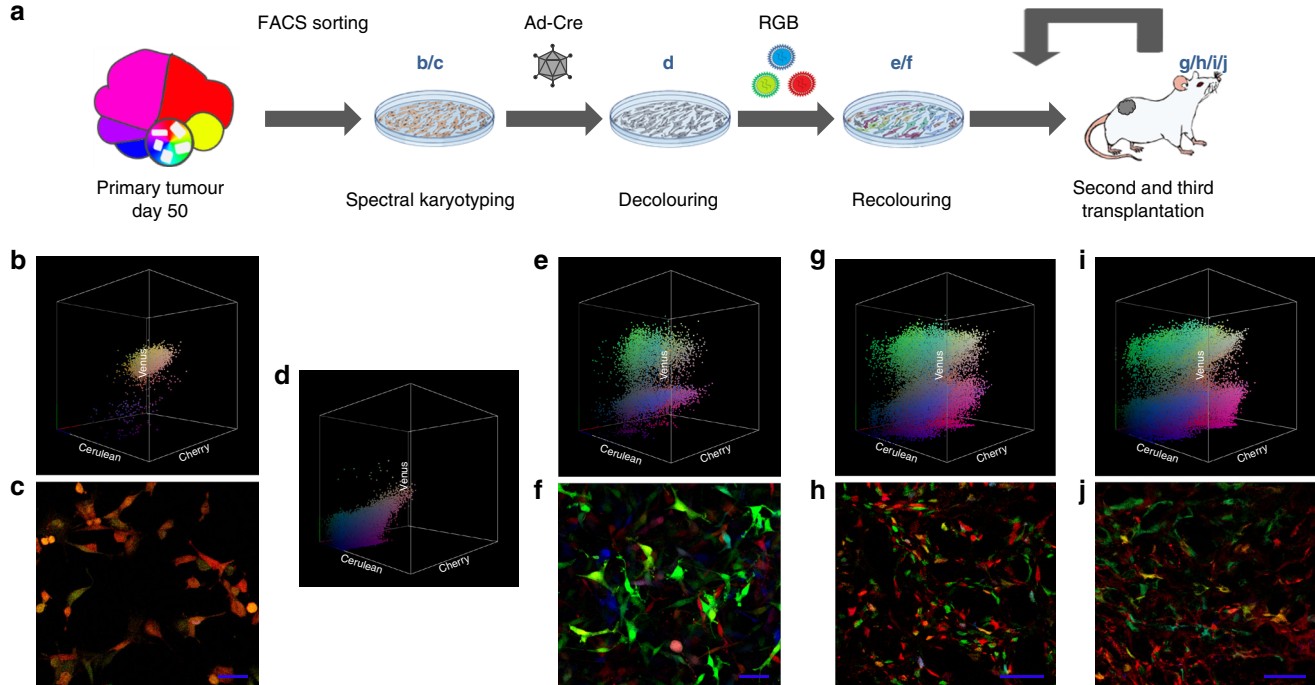

**Fig. 6** Isolated clones from RAINBONE tumours regenerate polyclonal tumours. **a** Experimental design of RAINBONE subclone recolouring and their in vivo tumourigenic potential (secondary $n = 9$, tertiary $n = 9$). The 3D flow cytometry analysis (**b**) and confocal image (**c**) of representative RAINBONE FACS sorted clones ($n = 3$). **d** Representative 3D flow cytometry analysis of decoloured clones after Ad-Cre transduction. Representative 3D flow cytometry analysis (**e**) and confocal microscopy image (**f**) of RGB-marked (recoloured) FACS sorted clones (RAINBONE-2). Representative 3D flow cytometry analysis (**g**) and confocal microscopy image (**h**) of RAINBONE-2 tumours (15 days of secondary transplantation) showing a polyclonal population. Representative 3D flow cytometry analysis (**i**) and confocal microscopy image (**j**) of RAINBONE-2 tumours (15 days after tertiary transplantation) again showing a polyclonal contribution. Blue bars = 50 μm

competitors and thus reducing cell–cell interaction[44]. Dispersal forces could also explain the metastatic process, a paradoxical outcome of tumour evolution that is not related to cell survival the way that other tumour hallmarks are (apoptosis, immune evasion, etc.)[45,46]. The main basis of the neutral theory is the neutral outcome of this process of competition, in which different species (cancer clones) mainly coexist and the acquisition of new genomic traits is mostly neutral. This model contrasts with the reiterative positive selection postulated by the Darwinian model of evolution. According to the Darwinian model, the acquisition of new phenotypic traits gives an increased replicative fitness to a new species with the continuous extinction of the unfit ones. Nevertheless, the neutral and competitive models are not completely in antithesis, and neutral evolution also allows positive selection[27,47]. The difference between the two models is mostly concerning the frequency of the positive selective events, which are defined as rare in the neutral theory. Positive selection is mostly relegated to strong microenvironmental changes, chemotherapy, immunotherapy, metastatic spread, and during the first stage of tumour evolution[27,48]. Therefore, after the accumulation of genomic alteration that initiates tumour growth, cancer cells expand neutrally and accumulate extensive genomic heterogeneity.

In line with the neutral model, within a short time of tumour evolution (25 days), OS tumours systemically presented histological heterogeneity together with a polyclonal distribution of tumour cells, thus not resembling a strongly selective linear model of clonal cancer evolution[16]. As in Nature, clonal competition is also occurring in our model, in which the dynamics of competition among cancer clones represent forces able to slow down tumour growth. Nevertheless, the outcome of this competition is not resulting in clonal selection. We observed that

different clones characterized by high genomic heterogeneity contributed with proliferating or differentiating cells in the tumour (Fig. 7a). This behaviour is in agreement with a neutral evolution pattern and lacks evidence of a real selective advantage gain. In contrast, at late-stage evolution (50 days), tumours showed a different organization; large extracompartmental monoclonal areas arose adjacent to the osteoinductive area generated in the hydroxyapatite/tricalcium phosphate (HA/TCP) compartment, presumably as a consequence of the effective pressure for clonal selection caused by different microenvironmental conditions. In this sense, clonal selection seems to be mostly associated with an adaptation to new specific spatial/microenvironmental determinants (Fig. 7a). It is important to consider that one parameter in the staging system of musculoskeletal tumours is the ability to grow extracompartmentally, which is also associated with a worse prognosis.

The fact that different clones within the same tumour are able to grow extracompartmentally provides evidence of a parallel evolution model among cancer cells. Additionally, in agreement with a contingency evolution context, our data suggest that starting from a pool of transformed cells, the chance of becoming a clone with this phenotype is not pre-established but that constraints can lead to convergence on this possible outcome. In fact, each specific tumour in each animal shows different dominant clones. Karyotype analysis of these clonal populations revealed extended karyotype variability among cells, which is compatible with a divergent pathway of cancer evolution and in accord with branched models. At the subclonal level, these cells show a high tumour-initiating potential, which is also in agreement with a secondary and tertiary level of coexistence (Fig. 7a). However, it is common to observe low-frequency clones infiltrating the dominant clonal population in these extracompartmental regions,

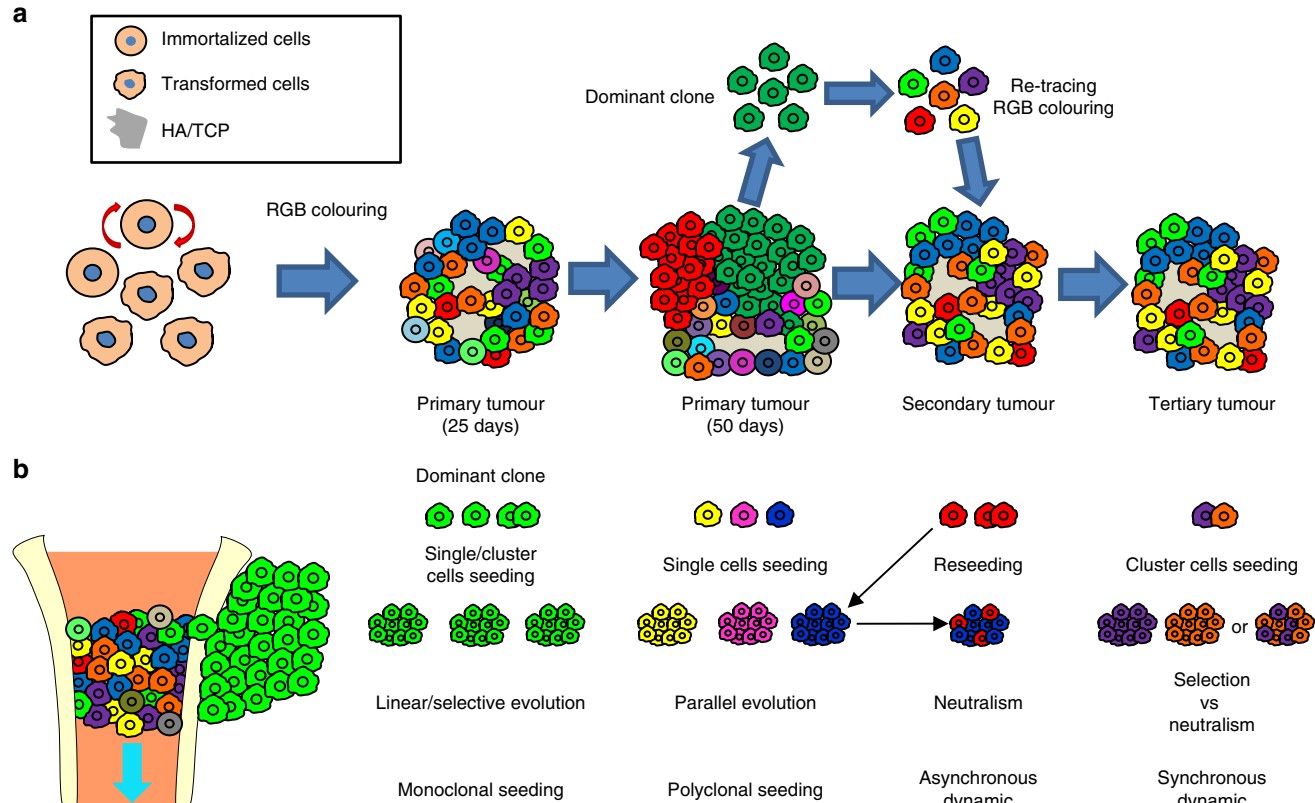

**Fig. 7** Schematic representation of the clonal dynamics in osteosarcomagenesis. **a** Schematic model of ectopic OS generated by transformed MPCs; the model indicates clonal heterogeneity over time; the first level of neutral evolution is typical of tumours at 25 days of development; clonal equilibrium is maintained until the development of different locally dominant clones in the extracompartmental region at 50 days. Dominant clones are sub-clonally heterogeneous and establish tumour development through a new level of neutral coexistence in secondary and tertiary transplantation. **b** Schematic model of OS developing at orthotopic location (left) and metastatic disease (right). Note the clonal heterogeneity in the medullary space and the outgrowth of locally dominant clones (one is depicted for simplification). On the right, the possible clonal dynamics of metastatic seeding are indicated; changes in colonization strategy (single cells vs. cluster), evolution models (selection vs. parallel vs. neutralism), clonality (monoclonality vs. polyclonality), and timing (synchronous vs. asynchronous) of the process are indicators of the possible clonal origin and the final clonal composition of metastatic nodules. Our model presents evidence of parallel evolution and polyclonal synchronous seeding dynamics

which adds a new level of heterogeneity. This type of pattern strongly resembles the dynamics of collaboration and/or parasitism that were proposed in other reports[49].

In summary, our study demonstrates that different dynamics simultaneously participate during tumour evolution, but we support the idea that the clinical relevance of tumour evolution should not be restricted only to dominant clones. In support of this concept, a clinical report in osteosarcoma described how some low-frequency clones detected at diagnosis could be responsible for tumour relapse, thus underlining the importance of low-frequency populations of cancer cells and the less fruitful branches of tumour evolution[50].

Metastatic seeding is another malignant feature of OS disease that can follow different dynamics. Some interesting reports, also employing multicolour lineage tracing, have started to highlight unknown mechanisms occurring during metastatic dissemination. Thus, the monoclonal or polyclonal dynamics of metastatic spread, collective dissemination, and reseeding were reported in different models of carcinomas[51–53]. Interestingly, some authors demonstrated that clonal cooperativity in cancer dissemination may play a primary role in improving the chances of engraftment at distant sites. In a model of pancreatic cancer, polyclonal clusters of cancer cells actively colonized distant organs, representing a cooperative strategy, and a reduction of cluster formation also reduced metastatic potential[45]. In our OS model, we

observed the development of different monochromatic and oligochromatic nodules, in accord with a polyclonal seeding (Fig. 7b), and a parallel progression model in lung colonization. Furthermore, the existence of oligoclonal metastases raises new questions about their origin and their intermetastatic subclonal dynamics because different clones can still coexist. Given the inefficiency of metastatic clones to home in pre-existing metastases, oligoclonal nodules do not seem to be the outcome of a secondary seeding wave, or reseeding (Fig. 7b). This result could be explained by the existence of a local microenvironment that allows the homing of monoclonal or oligoclonal seeds at the beginning of the disease and that impedes the engraftment of new clones once perturbed. In summary, our data question the competitive linear model in metastatic evolution and indicate that clonal dynamics occurring in metastatic disease do not differ from the dynamics at the primary site.

In conclusion, tumour evolution is thought to be caused by clonal competition and selection, which would lead to aggressive clone development in the fight for the survival of the fittest. By contrast, we present evidence that osteosarcomagenesis can follow the dynamics of neutral evolution, in which different cancer clones coexist and propagate simultaneously over time. Clonal biodiversity seems to be an important feature in our model. This equilibrium is maintained until the disease progresses to a more aggressive form that is associated with the invasion of an adjacent

microenvironment where dominant clones appear. Distant lung polyclonal seeding also results in the spatial dominance of many clones, which can be distinct from the dominant clones in the primary tumour. In summary, our study offers an overview of the clonal dynamics and relevance of dispersal forces in OS development; this knowledge is useful for understanding tumour biology and may improve clinical practice and therapeutic design.

## Methods

**Cell lines**. Murine BM-MPCs were isolated from transgenic FVB mice bearing *Tp53* and *Rb* genes flanked by LoxP sites. Gene deletion was achieved in vitro by the adenoviral transduction of Cre recombinase gene under the control of cytomegalovirus promoter to obtain transformed $p53^{-/-}Rb^{-/-}$ BM-MPCs. Successful gene knockdown was confirmed by genomic PCR and western blot[34]. $p53^{-/-}Rb^{-/-}$ BM-MPCs underwent lentiviral RGB marking in vitro. Cells were maintained in Dulbecco's modified Eagle's medium supplemented with 10% foetal bovine serum, 1% penicillin/streptomycin, and 1% Glutamax and were routinely tested for mycoplasma presence (MycoAlert-Mycoplasma Detection kit, LONZA).

**RGB lentiviral vectors and RGB multicolour marking**. LeGO-RGB lentiviral vectors were used as colour-guided clonal cell trackers. LeGO vectors were kindly provided by Dr. Kristoffer Riecken, University Medical Center Hamburg, Germany[36,37,54]. LeGO-Cer2 (Addgene: 27388), LeGO-V2 (Addgene: 27340), and LeGO-C2 (Addgene: 27339) plasmids were employed to produce lentiviral vectors coding for blue, green, and red fluorescent proteins, respectively. Supernatant was collected 48 h after transfection and concentrated by ultracentrifugation. Lentiviral particle mixtures were added to the $p53^{-/-}Rb^{-/-}$ BM-MPCs and incubated overnight to generate RGB multicolour-marked murine $p53^{-/-}Rb^{-/-}$ BM-MPCs, which were named RAINBONE cells. The RGB colour mix was achieved using a MOI of 0.75, which corresponds to an equimolar transduction efficiency of 50% per lentiviral vector 3 days after transduction. Six monoclonal cell lines were derived from RAINBONE cells by in vitro limiting dilutions. Single-cell plating efficiency and clonal purity were assessed by confocal fluorescence microscopy and flow cytometry, respectively. LeGO-RGB vectors also contain additional loxP sites which allow the elimination of fluorescent proteins using a Cre recombinase. Three different FACS sorted RAINBONE clones were decoloured by the in vitro adenoviral transduction of Cre recombinase. Decoloured clones underwent lentiviral recolouring. These recoloured clonal populations represent the RAINBONE-2 cells that underwent secondary and tertiary in vivo tumour generation.

**Flow cytometry analysis and cell sorting**. Cells were resuspended in phosphate-buffer saline (PBS) for flow cytometric study. Fluorescence signal distribution was analysed using a BD LSRFortessa (BD Bioscience) cell analyser. Cerulean fluorescent protein was excited at 405 nm and detected with a 450/50 bandpass filter, Venus was excited at 488 nm and detected with a 530/30 bandpass filter, and Cherry was excited at 561 nm and detected with a 610/20 bandpass filter. Discrete cell populations developed in osteogenic implants were further sorted using a iCyt SY3200 Cell Sorter (SONY). Cerulean fluorescent protein was excited at 405 nm and detected with a 525/50 bandpass filter, Venus was excited at 488 nm and detected with a 525/50 bandpass filter, and Cherry was excited at 532 nm and detected with a 615/30 bandpass filter. Sorted populations were expanded in vitro for a short period, and sorting purity was verified by flow cytometry and confocal microscopy. Flow cytometry data were analysed with FlowJO software (FlowJo LLC).

**Unsupervised visualization analysis of clonal architecture**. FCS files were loaded in the Cytobank website (https://premium.cytobank.org) to perform different types of unsupervised analysis of the clonal architecture of the samples. Samples were gated to analyse the mononuclear cell fraction. The clustering of sample events was performed, taking into account only the intensity of Cerulean, Venus, and Cherry channels. For this aim, a viSNE map was generated; this approach uses *t*-distributed stochastic neighbour embedding (t-SNE) algorithms[55]. The generated results are provided in two-dimensional scatter plots and show the intensity of the three fluorescent channels analysed. A SPADE algorithm was used to extract population hierarchies and visualize individual clones in a tree-like structure[56]. SPADE performs density-dependent down-sampling to equally represent rare and abundant populations and then performs agglomerative clustering while taking into account the intensity of selected channels. In this case, SPADE was used to cluster and represent the data as 200 different clones.

**Fluorescent microscopy analysis**. In vitro confocal microscopic studies of RAINBONE cells were performed by seeding cells in multichambers. After overnight incubation, slides were washed with PBS and fixed with 4% formalin or 1% paraformaldehyde (PFA) for 1 min. After fixation, slides were washed again with PBS and mounted with ProLong. In the case of explanted primary tumours and lungs, samples were processed for histologic staining and confocal fluorescence analysis by cryosectioning. Samples were fixed overnight in 4% formalin or 1% PFA

and decalcified for 72 h prior to inclusion in optimal cutting temperature Tissue-Tek. All processes were performed in the dark at room temperature. The 8 μm slides were defrosted and stained according to histologic standards or pre-warmed, hydrated in PBS for 2 min, and then mounted using ProLong for confocal microscopy studies. A confocal multispectral TCS-SP5 (Leica Microsystems) microscope was employed in this study. Representative images were obtained by maximum projection of a 10-layer stack of 8 μm-thick sections. Images were processed using LAS AF (Leica Microsystems). For lung seeding quantification, metastatic nodules were screened to identify the fluorescent fingerprint of i.v. inoculated FACS clones. A colocalization study was performed with ImageJ to ensure the presence of fluorescent markers specific to the inoculated clone and to exclude ambiguous cells. Macroscopic fluorescence and/or brightfield image maps were acquired with TCS-SP5 (Leica Microsystems) and AxioScan.Z1 (Zeiss). LAS AF and ZEN 2.3 (blue edition) were employed for image processing.

**Mouse models**. All procedures and animal care were performed at the National Institute of Health Carlos III (ISCIII) with the approval of the Institutional Animal Research and Welfare Ethics Committee according to the EU Directive for animal experiments in a specific pathogen-free environment. Experiments were performed using 8–10-week-old NSG mice. A minimum sample size of 4 mice per each experimental group was established; this size was chosen in accordance with 3 Rs (Replacement, Reduction, and Refinement) rule for animal experimentation, ensuring sufficient statistical power in dichotomous studies. Inclusion or exclusion criteria were pre-established and represent the physiological status of the animal at final experimental point. OS development was induced using two different procedures. For the orthotopic inoculation into the bone marrow space of the proximal tibia, cells were resuspended in PBS, filtered through a 70 μm nylon filter, and concentrated to $7.5 \times 10^6$ cells/ml. Surgery was performed by bending the mouse leg at 90° to drill the tip of the tibia with a 25 G needle and depositing the cell suspension in the medullar space ($1.5 \times 10^5$ cells/mouse) with a 27 G needle. For ectopic osteogenic implants, 40 mg of ceramic powder (60% hydroxyapatite/40% tricalcium phosphate beta) with a surface microporosity less than 10 μm (Biomatlante) was deposited in a 50-ml falcon tube and washed with 1 ml of culture medium. The cell suspension was mixed with ceramic powder ($1.5 \times 10^5$/implant), centrifuged at 1200 rpm for 5 min in a centrifuge with a swinging bucket rotor, and incubated overnight. Culture medium was carefully removed, and cells with ceramic powder were bound in a fibrin clot for subcutaneous implantation[34]. For bioluminescent studies, RAINBONE cells were transduced at MOI 5 with lentiviral particles carrying the firefly luciferase gene. Lentiviral vectors were produced using a phR-SIN-SFFV-pLuc-IRES-GFP transfer vector[57] after the deletion of GFP. This vector was employed to quantify tumour growth in vivo without affecting the RGB marking. In vivo bioluminescent quantification was performed using an IVIS Lumina3 image system (Perkin Elmer). Mice were anesthetized with 2% isoflurane and were imaged in ventral positions 1 min after the intravenous administration of 100 μl of a D-luciferin solution (12.5 mg/ml in PBS). Data were analysed using Living Image software (Perkin Elmer). Quantification was performed by subtracting mouse background average radiance from the ROI average radiance of primary tumours and thorax. For subcutaneous in vivo growth kinetic studies, normalization was calculated as fold change to day 1 after implantation or by the basal cell line luminescence (SCLL) measured in vitro. SCLL was calculated as luciferase activity per μg of protein. For orthotopic studies, the normalization factor is represented by the average radiance 4 h post inoculation. For animal experimentation, researchers were not blind; researchers involved in flow cytometry data acquisition and LAM-PCR analysis were blind.

**SKY analysis**. RAINBONE cells and explanted tumour cells underwent molecular cytogenetic analysis. Cultured adherent cells were treated with colchicine (0.5 μg/ml) for 4 h at 37 °C and routinely harvested. Metaphases were prepared using a conventional cytogenetic protocol for methanol/acetic acid (3:1)-fixed cells. Slides were prepared from the fixed material and hybridized using the SKY method according to the manufacturer's protocol (Applied Spectral Imaging). Images were acquired with an SD300 Spectra Cube (Applied Spectral Imaging) mounted on an Axioplan microscope (Zeiss) using a custom-designed optical filter, SKY-1 (Chroma Technology). Up to 15 metaphase cells were captured and analysed for each cell line when possible.

**Genomic insertion site analysis by LAM-PCR**. Lentiviral integration site analysis was performed by a modified LAM-PCR[58,59]. This method amplifies the DNA around the viral/host junction and identifies the adjacent host DNA by sequencing. A single-stranded copy of the proviral-host junction was made by linear extension from a biotinylated viral LTR primer; this single-stranded junction fragment was trapped and isolated on streptavidin-coated magnetic beads, and a second strand was generated by random-primed polymerization. The host sequence was cut at the nearest *Rsa*I or *Hea*III restriction site and ligated to an anchor primer; the junction region was amplified by nested PCR. The complexity of the population of integration sites was initially monitored by running PCR products on 4–20% polyacrylamide gel. Individual insertion sites were identified by cloning and sequencing individual bands. The non-viral sequence from each band was used to search the mouse genome using a BLAST search on the University of California Santa Cruz

Genome Browser. Uncropped polyacrylamide gels images are presented in Supplementary Figure 12.

**Statistical analysis**. Data were graphed with GraphPad Prism (GraphPad Software) and Excel service pack (Microsoft software). Statistical analyses were performed by GraphPad Prism (GraphPad Software). Data are expressed as the means ± SD unless otherwise specified. In boxplots graphs, centre line indicates median, bounds of box indicate 25th and 75th percentiles, and whiskers indicate minimum and maximum. Correlation between two parameters was estimated by Pearson's coefficient of correlation, by two-tailed tests, and with a confidence interval of 95%. $P$ values less than 0.05 were considered statistically significant. One-way or two-way analysis of variance (ANOVA) with Bonferroni post-testing was used to compare significant differences for more than two groups. For multiple comparisons, a confidence interval of 95% was adopted, and only $P$ values lower than 0.05 were considered statistically significant. $*P < 0.05$, $**P < 0.01$, $***P < 0.001$, and $****P < 0.0001$ were deemed statistically significant.

## Data availability

Datasets generated and analysed in the study are available from the corresponding author upon reasonable request.

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

## Acknowledgements

We thank ISCIII and CNIO flow cytometry and cell sorting units for their participation in our studies. We are thankful to the CCEH-Fred Hutchinson Cancer Research Center for LAM-PCR service. We acknowledge Raquel Pérez Tavarez, María Blázquez Mesa, Alicia Giménez Sánchez, Elena Calvo Cazalilla, and Monserrat Arroyo Correas for useful help on the pathology studies; and Teresa Cejalvo, Isabel Cubillo Moreno, and Miguel Angel Rodríguez-Milla for their contributions in experimental setup. We thank the visual artist Isabella Lacquaniti for her help with drawings and schematics. We are also thankful to the Fondo de Investigaciones Sanitarias (FIS: PI11/00377 and PI14CIII/00005 to J.G.-C., FIS: CP11/00206 to A.A., and RTICC: RD12/0036/0027 to J.G.-C.), the Madrid Regional Government (CellCAM; P2010/BMD-2420 to J.G.-C.), the Asociación Pablo Ugarte, and the Asociación Afanion for grants support.

## Author contributions

Conception and design: S.G., A.A. and J.G.-C. Development of methodology: S.G., A.A. and J.G.-C. Acquisition of data: S.G., F.G., and A.M.-M. Analysis and interpretation of data (e.g., statistical analysis, biostatistics, computational analysis): S.G., F.G., A.M.-M., A.A. and J.G.-C. Writing, review, and/or revision of the manuscript: S.G., A.A., A.M.-M., J.R., F.G., A.A. and J.G.-C. Administrative, technical, or material support (i.e., reporting or organizing data, constructing database): S.G., A.M.-M., J.R., F.G., A.A. and J.G.-C.

## Additional information

**Competing interests:** The authors declare no competing interests.

