## [Peer Review File · Nature Communications]

Reviewers' comments:

Reviewer #1 (Remarks to the Author):

The major claims of this paper are that evolution of polyclonal tumors occurs in parallel in osteosarcoma, and that even metastases at times include oligoclonality. All of this is based on a model that has a number of strengths, but also weaknesses. Mesenchymal progenitor cells with homozygous floxed alleles for Trp53 and Rb1 are transformed in vitro by administration of AdCre, followed by the application of lentiviral delivery of RGB to generate multi-colored specific clones. Most of the data presented are examples from small sample size experiments that demonstrate maintenance of multiple color clones in both in vitro or short-term in vivo growth, with more clonal areas demonstrable in the extracompartamental outgrowths.

The major problem conceptually is that not every mesenchymal progenitor cell that loses Trp53 and Rb1 truly transforms into an osteosarcoma. In mouse genetic models of osteosarcomagenesis, there are typically only a few tumors that form even after entire mesenchymal lineages lose both alleles of both of these genes. Such genetic manipulations, however, are not only either transforming or lethal to cells, but may render cells fairly happily immortal, even if not malignant. Because of this, there will, of course, be parallel clones that persist, not because of parallel development of many tumors, but merely by parallel survival of non-transformed cells in, around, and among the few actually transformed clones. The few tumor clones that actually develop the ability to grow beyond the initial compartment are likely the real tumors, and these are much more clonal, almost arguing against their claim.

The appearance of multiple clones in metastases is more interesting. Some numbers regarding how many of each type of oligoclonal metastasis would be helpful. The example shown in Fig5D, for example, may merely be two adjacent metastases, rather than one metastasis from two clones. How many were of each type?

Figure 7 mostly shows that on-plastic growth promotes instability more than in vivo growth. Of course, that may be due to the fact that these cells are selected more powerfully in that harsh environment. It is certainly not a novel finding in and of itself.

Reviewer #2 (Remarks to the Author):

Gambera et al applied an individual-cell colouring technique named RGB marking to study clonal dynamics of primary tumours as well as metastases in a murine osteosarcoma (OS) model. Generally, despite inherent technical limitation noted by the authors, this is a very interesting approach that may provide novel insights in tumour heterogeneity and processes of intra-tumour competition. Overall this is a well-designed study containing interesting data. However, before publication the authors in my opinion need to address several weaknesses of their work.

Major points

1) The main conclusion Gambera et al draw from their work is that "osteosarcomagenesis can follow a dynamic of neutral evolution". However, I propose that it is important to take into consideration a relevant limitation of the used model. In fact, in the used model of OS development is based on the simultaneous knockout of two major regulator genes, namely p53 and Rb. Such double-hit is quite far from the natural evolution of a tumour, so it might not be surprising that by using this approach the authors generated multiple cells with potentially equal tumour-initiating capacity. Indeed, the in-vivo SKY data indicates that the tumour cells do not undergo major clonal evolution during engraftment and outgrowth (supposedly, since all participating clones are already quite malignant). Thus, the model might not be best suited to assess clonal competition. I won't say that the model cannot be used (each model has its strengths and weaknesses), but I propose that the authors need to more critically reflect this point in their data interpretation.

2) In their final in-vivo experiment (p. 12 and Fig. 6), the authors use a very elegant approach to decolour and recolour dominant clones using the Cre/lox system. Since most of their clones obviously contain several vector insertion sites, it would be interesting to know how efficient their vector excision was. In this context, it remains unclear, whether Fig. 6D shows decoloured cells after a sorting step, or the whole decoloured population after Cre expression. A potential caveat of expressing Cre in the presence of multiple lox sites might be the induction of chromosomal rearrangements potentially impacting the readout. Did the authors include Rainbone II cells in their SKY analyses?

3) The Fluorescence microscopy (FM) data (e.g. in Fig. 1A, 2D, E, Suppl. Figure 1) mostly show cells of the basic colours red, green and blue. This is in contrast to the Flow cytometry (FC) data indicating efficient marking with all three vectors that should, following the RGB principle, result in many mixed colours. The apparent contradiction might be due to the modern data processing in FM, where weak signals are often set to black to improve contrast. The authors might want to check if this was the reason for the underrepresentation of clones with mixed colours in their FM pictures.

4) Fig. 4: B/C: It is difficult to allocate the section shown in 4C to the main picture in 4B. Could the authors put a box on the respective region in Fig. 4B. F/G: Are these regions really clonal? From the photographs this does not become convincingly clear.

5) Results (p. 10 and Fig. 5): Application of in-vivo imaging comes quite as a surprise for the reader, since marking with the luciferase gene had not been introduced at any point before (later on there are two lines in the Method section). Obviously, the authors need to introduce this additional experimental setting to make their results interpretable.

Minor points:

1) Fig 1D (referred to on page 9) is missing. It has obviously become Suppl. Fig. 3C instead.

2) The actual value of the LAM-PCR, e.g. in Fig. 2F remains a bit odd.

3) The manuscript would profit from editing by a native speaker, it contains quite some grammatical errors and misprints.

4) The authors mention that the used vectors contain lox sites (s. also major point 2). However, no vector map is provided to estimate what part of the vector is deleted after Cre application. Could the authors provide vector maps or at least a reference to a paper containing those maps? For the readers it would also be useful to know whether the vectors are available from Addgene.

5) p. 23 and Fig 3G: the restriction enzyme was probably HaeIII (as also said in the Table of Fig. 3G), not HeaIII?

6) Did the authors also perform SKY analysis on the parental cells for comparison?

Reviewer #1 (Remarks to the Author):

“The major claims of this paper are that evolution of polyclonal tumours occurs in parallel in osteosarcoma, and that even metastases at times include oligoclonality. All of this is based on a model that has a number of strengths, but also weaknesses. Mesenchymal progenitor cells with homozygous floxed alleles for Trp53 and Rb1 are transformed in vitro by administration of AdCre, followed by the application of lentiviral delivery of RGB to generate multi-colored specific clones. Most of the data presented are examples from small sample size experiments that demonstrate maintenance of multiple colour clones in both in vitro or short-term in vivo growth, with more clonal areas demonstrable in the extracompartmental outgrowths.”

We agree with Reviewer 1. A relevant conclusion can be obtained with a significant sample size, which is obviously needed to fully assess the experimental outcome. Nevertheless, according to the three Rs rule (Replacement, Reduction, and Refinement) for animal studies, an excessive sample size can result in a waste of animal life because equally valid information can be obtained from a small number of subjects. In our study, even if a quantitative approach is taken, our hypothesis and results mainly fall into two categories: monoclonality and oligoclonality (or polyclonality). This categorization corresponds to the grade of differences present in a dichotomy. Given the unknown incidence of monoclonal or polyclonal osteosarcoma in nature, we assumed an incidence of 50% among the general population, and with only 4-10 mice, we can test extremely different experimental outcomes, such as monoclonality and polyclonality (false positive rate of 1-10%, false negative rate of 95%) (**Letter Figure 1**). Larger experimental groups were also employed to ensure technical and biological replicates for data analysis (**Letter Table 1**).

Letter Figure 1: Sample size calculation for dichotomous experiments. The figure indicates the calculation of the minimum number of subjects (N) to have sufficient statistical power in dichotomous studies⁷. p_0 = proportion (incidence) of population; p_1 = proportion (incidence) of study group; α = probability of type I error; β = probability of type II error; z = critical Z value for a given α or β .

Experiment	Conditions	Number of Subjects (N)
Ectopic (bone-like)	25 & 50 days	21
Retracing	3 clones	20
Orthotopic	50 days	10
Reseeding	2 clones	6
Clonal Competition	Limiting dilution & Dominant Clones	20

Letter Table 1: *In vivo* experiments grouped for category and sample size employed (N).

Major Points

“The major problem conceptually is that not every mesenchymal progenitor cell that loses Trp53 and Rb1 truly transforms into an osteosarcoma. In mouse genetic models of osteosarcomagenesis, there are typically only a few tumors that form even after entire mesenchymal lineages lose both alleles of both of these genes. Such genetic manipulations, however, are not only either transforming or lethal to cells, but may render cells fairly happily immortal, even if not malignant. Because of this, there will, of course, be parallel clones that persist, not because of parallel development of many tumors, but merely by parallel survival of non-transformed cells in, around, and among the few actually transformed clones. The few tumor clones that actually develop the ability to grow beyond the initial compartment are likely the real tumors, and these are much more clonal, almost arguing against their claim.”

We agree with Reviewer 1 that, theoretically, our approach would generate a pool of cells in which not all are truly transformed. To characterize the tumour-initiating potential of RAINBONE cells after RGB colouring, we generated monoclonal populations obtained by *in vitro* limiting dilution of the original RAINBONE pool of cells. We compared the *in vivo* growth dynamics of these monoclonal populations individually and in mixtures. We prepared a clonal mixture of 5 or 10 different monoclonal populations, and we also tested the original RAINBONE cells, which represent the most complex mixture (**Letter Figure 2-Left/new Supplementary Fig. 5**). Our results indicate that all of the monoclonal populations tested have TIC properties (4 out of 4) and that the tumours generated by mixed population grow slowly *in vivo*, most likely due to clone interactions.

Letter Figure 2: Increasing clonal complexity reduces tumour growth.

A, Bioluminescent *in vivo* growth quantification of tumours generated by monoclonal, oligoclonal, and polyclonal populations during 25 days of *in vivo* growth in NSG mice (two different experiments, n=24). Seven monoclonal populations were derived from *in vitro* limiting dilution clones (single clones; n=10) of RAINBONE cells; oligoclonal mixes represent cell lines obtained by mixing 5 (clonal MIX5; n=6) or 10 (clonal MIX10; n=2) clones of the previous monoclonal populations; RAINBONE cells represents tumours originated by the parental cell line (n=6). The upper graph shows representative luciferase activity of the different experimental groups in one experiment. Graphs located in the middle of the panel show normalized luciferase activity of two experiments, each graph represent a different one; normalization is performed by calculating the fold change to the respective average radiance at day 1 of *in vivo* growth (left panel) or normalized against the specific cell line luminescence (SCLL) of tumour cells *in vitro* (right panel). In the lower part of the panel is represented the growth kinetic of different single clones versus clonal MIXs, each graph represent a different experiment. The panel show a tendency of slower tumour growth at increasing clonal complexity and heterogeneous growth kinetics for tumours generated by monoclonal populations. **B,** graph show the percentage of colonies formed *in vitro* by RAINBONE cells in soft agar colony formation assay (n=8) where two different seeding density were tested (100 or 200 cells/per well); error bars represent standard deviation. In the right part of the panel, representative images of RAINBONE colonies versus wild type murine MSCs, which do not form colonies in semi-solid medium. White bars= 100 μm. **C,** Bioluminescent *in vivo* growth quantification of tumours generated by monoclonal, oligoclonal, and polyclonal populations during 25 days of secondary transplantation in NSG mice (n=10). Four monoclonal cell lines (single dominant clones; n=4) are derived by FACS-sorted dominant clones isolated from primary tumours (clonal origin is defined later in the text); oligoclonal mix represents a population obtained by mixing the previous 4 dominant

clones (clonal MIX4; n=4); RAINBONE cells were implanted as control cells, which was their first time *in vivo* (n=2). The upper graph of the panel shows luminescence quantification of the three experimental groups. Luminescence normalization is calculated against the specific cell line luminescence of explanted tumour cells *in vitro* (SCLL). In this case also, the graphs show a tendency toward slower tumour growth at increasing clonal complexity (central graph in the panel) and an heterogeneous growth kinetics for tumours generated by monoclonal populations (lower graph in the panel). **D**, the left panel shows the clonal composition of two ClonalMIX4 populations after 25 days of secondary *in vivo* growth as analysed by flow cytometry; G11, R7, R9, and R11 represent the 4 dominant clones. The right panel shows *in vitro* growth assays for dominant clones; each clone was grown separately (n=3). G11 (pink arrow) and R7 (red arrow) *in vivo* tumour growth kinetics are also indicated in Panel B. All data in the picture are presented as single values or as the means and standard deviations; for box and whiskers plots, means plus minimum and maximum values are presented; statistic tests: two-tail unpaired t-Test and two-way ANOVA, Bonferroni post hoc test, CI: 95%, alpha: 0.05. P<0.05 (*), P<0.01 (**), P<0.001 (***), P<0.0001 (****).

Different studies have addressed the tumour-initiating potential of mesenchymal lineage cells, and we agree with Reviewer 1 that each model has its strengths and weakness. Most of the published studies have tested tumour-initiating potential using harsh *in vivo* assay conditions (ectopic inoculation) and, depending on the study, more or less immunocompetent mice. As shown in other cancer models, tumour-initiating cell assays can be improved if cells are implanted in more immunodeficient mice and/or if embedded into a matrix at implantation⁸. An added value of our work is the successful adoption of a method to test cellular transformation under osteogenic conditions in NSG-immunodeficient mice; this may be the priming event needed for mesenchymal stem cell models of osteosarcomagenesis⁹. Using this model, and as Review 1 indicated, it is also reasonable that the tumour-initiating potential is not the same among all cells (**Letter Figure 3 Panel A/new Figure 7**). However, Reviewer 2 proposed the opposite point of view that all cells would have the same tumour-initiating potential (Letter Figure 3, Panel B). We would like also to highlight that our work is structured at different time points with clones coexisting and following neutral dynamics in secondary and tertiary transplants.

Letter Figure 3: Schematic view of Reviewers' comments and neutral evolution levels.

In any case, we support the Reviewers' interpretation of a heterogeneous composition, but we do not exclude that tumour cells (transformed and/or immortal) actually represent fixed entities in the tumour. Indeed, tumorigenic potential could require more time (cell doubling) or different microenvironmental conditions (extracompartmental constraints, chemotherapy-induced mutations, or carcinogens) to establish immortal cells (Letter Figure 3, Panel A). This assumption is extremely important because in the absence of ongoing clonal selection, these immortal clones would be dormant cells in the primary tumour. Our model further enforces the concept of targeting therapies against major genetic alterations (even if pre-cancerous) and not only against the events responsible for extracompartmental growth.

“The appearance of multiple clones in metastases is more interesting. Some numbers regarding how many of each type of oligoclonal metastasis would be helpful. The example shown in Fig5D, for example, may merely be two adjacent metastases, rather than one metastasis from two clones. How many were of each type?”

We agree with Review 1; the metastasis data are interesting, and the unknown dynamics of seeding requires more attention and some quantification. Therefore, we have included the quantification of 146 lung nodules from 6 mice, which present a monoclonal and oligoclonal composition. These data are in the new version of Figure 5 (Letter Figure 4). As Reviewer 1 indicated, the previous Fig. 5D could be confusing; for this reason, we further quantified oligoclonal metastasis using more rigid criteria (**Letter Figure 4-Panel C**). We assumed that only nodules with an intermixed clonal (colour) composition could be considered as properly oligoclonal. Our results (**Letter Figure 4-Panel D**) indicate that monoclonal seeding is apparently more frequent than oligoclonal seeding (* $p < 0.05$). We further divided our statistical analysis according to metastasis size because the presence of single or small cell clusters (<200 μm) would not be indicative of effective cell engraftment. Our results indicate no significant differences in the frequency of monoclonal and oligoclonal nodules, independently of their sizes.

Letter Figure 4: C, Representative images of oligoclonal and monoclonal nodules and size heterogeneity; 146 nodules were quantified, and the results are presented in D, lung nodule quantification according to monoclonality and polyclonality (left graph, p-value= 0.0497) and further categorized by size (right graph, p-values not significant). Statistical tests: two-tail paired t-Test and two-way ANOVA, Bonferroni post hoc test, CI: 95%, alpha: 0.05. $P < 0.05$ (*). Brown bar = 500 μm; white bar = 200 μm.

Furthermore, we have three major findings concerning metastatic disease:

- 1) Metastatic clones do not always correspond to dominant clones at primary sites
- 2) Different clones can seed lungs
- 3) Metastatic nodules can be formed by oligoclonal clusters

These three conclusions implicate different important concepts about the clonal dynamics occurring in metastatic disease (Letter Figure 5, new Figure 7). Concerning the first and second findings, a pattern of polyclonal seeding with the existence of clonally distinct metastatic nodules is evidence of the parallel progression of aggressive features among cancer clones. Therefore, we exclude the linear and selective evolution model in this process. The third finding brings the discussion to the clonal dynamics occurring within a single nodule (selection/neutralism). The development of oligoclonal nodules further represents another level of clonal coexistence (see the previous discussion) and seems to be occurring in the initial steps of lung seeding. Accordingly, metastatic clones are inefficient at reseeding pre-existing metastases.

Letter Figure 5: Possible clonal dynamics occurring during metastatic disease.

In total, these three concepts could have also important therapeutic implications; metastatic cells could be more resistant to chemotherapy treatment due to their clonal heterogeneity. Moreover, the differences between a primary tumour and metastatic cells may not be discernible from studying the dominant clones at the primary site or from a single biopsy.

“Figure 7 mostly shows that on-plastic growth promotes instability more than in vivo growth. Of course, that may be because these cells are selected more powerfully in that harsh environment. It is certainly not a novel finding in and of itself.”

We completely agree with Reviewer 1, and we also believe that karyotype instability on plastic is not a novel finding. Regrettably, this was not the point we wanted to make with our data; we apologize for the confusing presentation. The aim of this analysis was to determine the chromosomal instability (CIN) and genomic variability among RAINBONE cells and RAINBONE-derived tumour cells. This study showed us the abundant genomic heterogeneity at different stages of development, which further drove us to the following two main conclusions (Letter Fig. 3): in vitro, RAINBONE cells are genomically heterogeneous (starting point), and in vivo, dominant clones are genomically heterogeneous (end-point). These data also support the existence of extremely branching evolution, which further increases disease heterogeneity, and the data support the idea of sub-clonal coexistence among genomically distinct cancer clones.

In the new version of the manuscript, we have moved the old Figure 7 to a new Supplementary Figure 8, including a new Supplementary Table 1. The new data include a visual heatmap of the composite karyotype that was not presented previously to offer to the reader an overview of the dynamics of genomic heterogeneity in our study and to better explain our conclusions.

Reviewer #2 (Remarks to the Author):

“Gambera et al applied an individual-cell colouring technique named RGB marking to study clonal dynamics of primary tumours as well as metastases in a murine osteosarcoma (OS) model. Generally, despite inherent technical limitation noted by the authors, this is a very interesting approach that may provide novel insights in tumour heterogeneity and processes of intra-tumour competition. Overall this is a well-designed study containing interesting data. However, before publication the authors in my opinion need to address several weaknesses of their work.”

We wish to thank Reviewer 2 very much for the appreciation; the criticisms expressed are pertinent, and we found them to be good starting points to increase the robustness of our work. We hope to address and limit all the weaknesses indicated.

“Major point:

1) The main conclusion Gambera et al draw from their work is that “osteosarcomagenesis can follow a dynamic of neutral evolution”. However, I propose that it is important to take into consideration a relevant limitation of the used model. In fact, in the used model of OS development is based on the simultaneous knockout of two major regulator genes, namely p53 and Rb. Such double-hit is quite far from the natural evolution of a tumour, so it might not be surprising that by using this approach the authors generated multiple cells with potentially equal tumour-initiating capacity. Indeed, the in-vivo SKY data indicates that the tumour cells do not undergo major clonal evolution during engraftment and outgrowth (supposedly, since all participating clones are already quite malignant). Thus, the model might not be best suited to assess clonal competition. I won't say that the model cannot be used (each model has its strengths and weaknesses), but I propose that the authors need to more critically reflect this point in their data interpretation.”

In evolutionary theories, competition is a long and steady principle that is continuously occurring in ecological systems such as cancer^{10,11}. From an early stage of cancer development, tumour cells compete for limited resources (nutrients and oxygen) to the point of saturation¹² and encounter a strongly selective microenvironment (pH changes, immune system, geographic barriers, and chemotherapy), which is another component limiting their development¹³. In this context, dispersal forces may also play a key role in tumour progression¹⁴; populations of spreading cells could reach untapped resources, increasing the distance from competitors and thus reducing cell-cell interaction¹⁵. Dispersal forces could also explain the metastatic process, a paradoxical outcome of tumour evolution that is not related to cell survival the way that other tumour hallmarks are (apoptosis, immune evasion, etc.)^{16,17}. The main basis of the neutral theory is the neutral outcome of this process of competition, in which different species (cancer clones) mainly coexist and the acquisition of new genomic traits is mostly neutral. This model contrasts with the reiterative positive selection postulated by the Darwinian model of evolution. According to the Darwinian model, the acquisition of new phenotypic traits provides increased replicative fitness to a new species, which causes the continuous extinction of the unfit species. Nevertheless, the neutral and competitive models are not completely in antithesis, and neutral evolution also allows positive selection^{18,19}. The difference between the two models is mostly concerning the frequency of the selective events, which are defined as rare in the neutral theory. Positive selection is mostly relegated to strong microenvironmental changes, chemotherapy, immunotherapy, metastatic spread, and during the first stage of tumour evolution^{19,20}. Therefore, after the

accumulation of genomic alterations that initiate tumour growth, cancer cells expand neutrally and accumulate extensive genomic heterogeneity.

Our model is based in the double hits of p53 and Rb, which are frequent genomic alterations observed in osteosarcomas (20% to 80% of cases, depending on the study cohort), although the disease molecular pathogenesis is mostly unknown; nevertheless the double-hit approach offers us the ability to achieve the transforming events. As shown in Letter Fig. 2 (new Supplementary Fig. 5), tumour-initiating potential is a common feature of RAINBONE cells (4/4 of limiting dilution clones; 100% incidence); each monoclonal tumour shows different dynamics of *in vivo* growth. Furthermore, tumour clones compete in their primary microenvironment, and tumours show a tendency to develop faster at reducing clonal complexity (Letter Fig. 2 Panel A). Therefore, we believe that RAINBONE cells represent a good model to test neutral evolution theories because, as introduced previously, neutral models do not imply the absence of competition. Furthermore, RAINBONE cells at early stages of tumour evolution do not grow extracompartamentally, and thus they are a good model to assay the relevance of space limitation and dispersal forces. Once some clones acquired this phenotypic trait, they continued to diverge and acquire further genomic variability, as evidenced by the SKY analysis of extracompartamental clones. After this selective event, extracompartamental clones continue to compete in their development (Letter Figure 2 Panel B), but in agreement with a neutral model, different sub-clones can coexist (Letter Figure 2 Panel C and new Supplementary Fig. 8).

In any case, we agree with Reviewer 2 that the paper will benefit from this specific discussion, which is included in the new version of the paper (pages 15-17) (new Figure 7). We have also changed some sentences in the conclusions (page 15 and 17).

“2) In their final in-vivo experiment (p. 12 and Fig. 6), the authors use a very elegant approach to decolour and recolour dominant clones using the Cre/lox system. Since most of their clones obviously contain several vector insertion sites, it would be interesting to know how efficient their vector excision was. In this context, it remains unclear, whether Fig. 6D shows decoloured cells after a sorting step, or the whole decoloured population after Cre expression. A potential caveat of expressing Cre in the presence of multiple lox sites might be the induction of chromosomal rearrangements potentially impacting the readout. Did the authors include Rainbone II cells in their SKY analyses?”

The decolouring step was performed after SKY analyses, and thus the data represent the instability score of tumour clones at 50 days of development, just before decolouring and recolouring steps.

RGB-vector excision was verified by flow cytometry as shown in Fig. 6D, which represent the whole population of Fig. 6C after decolouring. No further FACS sorting was needed given the efficiency of Ad-Cre vectors (New Supplementary Fig. 10). Nevertheless, we do not exclude the possibility of Ad-Cre failure; it is possible that a LoxP site is corrupted and vector integration is resistant to Cre recombinase. We did not observe this event in our study; our clones were not resistant. We further tested decolouring efficiency by confocal microscopy, but obviously we did not present these data because they are black images. A more accurate quantification was achieved by flow cytometry, corresponding to a high efficiency of Ad-Cre decolouring, as presented in our representative image (Fig. 6D).

We apologize for the confusing data presentation; we have modified the Fig. 6A schematic and added a better explanation in the main text and figure legend to clarify the experimental design and Ad-Cre efficiency.

“3) The Fluorescence microscopy (FM) data (e.g., in Fig. 1A, 2D, E, Suppl. Figure 1) mostly show cells of the basic colours red, green and blue. This is in contrast to the Flow cytometry (FC) data indicating efficient marking with all three vectors that should, following the RGB principle, result in many mixed colours. The apparent contradiction might be due to the modern data processing in FM, where weak signals are often set to black to improve contrast. The authors might want to check if this was the reason for the underrepresentation of clones with mixed colours in their FM pictures.”

We agree with Reviewer 2; we should disclose that image acquisition was performed by adjusting laser intensities with fluorescent control tissue as background. Furthermore, the technique is designed to not saturate fluorescent signal. Nevertheless, due to some really weak signals, the results are completely appreciable only on large screens located in a dark room with the ability to zoom in and out in large image maps. We know that the total colour variability is not reproducible on projectors and printed paper, and thus we decided to increase the contrast only for data presentation and to increase the visibility of weak signals. This operation generated a flattening of the highest signals, colour saturation, and a reduction in the total colour tones, and thus this was only performed for image presentation. Another important bias that we want to recognize is represented by the human eyes, which tend to focus on the most intense signals; colour discrimination is even more complex when many colour units are mixed, as in the case of our cells (Fig. 2E). Some examples of the limitation of the human eye are presented in **Letter Fig. 6**. For this reason, we included a picture series in which we show red, green, and blue clones and some of their colour combinations (Fig. 2D). Knowing these limitations, the study also included flow cytometry, which more accurately quantifies each component of the fluorescent signal.

Letter Figure 6: Examples of limitations of the human eye in colour quantification and computer graphics artefacts. In both images, the strong green tone captures our attention when zoomed out (left), and the dispersal distribution (right) makes our colour frequency estimation difficult. All images have the same number of dots of 7 different colours.

“4) Fig. 4: B/C: It is difficult to allocate the section shown in 4C to the main picture in 4B. Could the authors put a box on the respective region in Fig. 4B. F/G: Are these regions really clonal? From the photographs this does not become convincingly clear.”

We apologize for the confusing data presentation. We prepared a new version of Fig. 4 that includes the box on the respective region, a modified figure legend, and reference text.

“5) Results (p. 10 and Fig. 5): Application of in-vivo imaging comes quite as a surprise for the reader, since marking with the luciferase gene had not been introduced at any point before (later on there are two lines in the Method section). Obviously, the authors need to introduce this additional experimental setting to make their results interpretable.”

We apologize for the missing information; we appropriately introduced all the experimental details in the text of the paper and added a new Supplementary Fig. 4.

*“Minor points:
1) Fig 1D (referred to on page 9) is missing. It has obviously become Suppl. Fig. 3C instead.”*

We apologize for the mistake; Fig. 1D does not exist. The corresponding SPADE analysis is included in Suppl. Video S1.

Suppl. Fig. 3C is the SPADE comparison of RAINBONE tumours at 25 days (left) versus RAINBONE tumours at 50 days (right). We modified the reference text.

“2) The actual value of the LAM-PCR, e.g., in Fig. 2F remains a bit odd.”

We agree with the comment of Reviewer 2; the image is a bit odd, most likely due to enhanced contrast. We needed to perform this image processing because some bands showed very weak signal and were not visible at the final image size. The same image processing of Fig. 2F was performed on Fig. 3F. In the new version of the paper, we added a larger image at higher resolution/size to give the reader a better view of this complementary result (Supplementary Fig. 7 in the new paper version).

“3) The manuscript would profit from editing by a native speaker, it contains quite some grammatical errors and misprints.”

We apologize for the grammatical errors and misprints. The new paper version was edited by an editing service (Springer Nature-Author Services; see below).

Nature Research Editing Service Certification

This is to certify that the manuscript titled *Clonal dynamics in osteosarcoma defined by RGB-marking*, was edited for English language usage, grammar, spelling and punctuation by one or more native English-speaking editors at Nature Research Editing Service. The editors focused on correcting improper language and rephrasing awkward sentences, using their scientific training to point out passages that were confusing or vague. Every effort has been made to ensure that neither the research content nor the authors' intentions were altered in any way during the editing process.

Documents receiving this certification should be English-ready for publication; however, please note that the author has the ability to accept or reject our suggestions and changes. To verify the final edited version, please visit our verification page. If you have any questions or concerns over this edited document, please contact Nature Research Editing Service at support@as.springernature.com.

Manuscript title: Clonal dynamics in osteosarcoma defined by RGB-marking.
Authors: Stefano Gambera, Ander Abarrategi, Fernando González-Camacho, Álvaro Morales-Molina, Josep Roma, Arantazu Alfranca, Javier Garcia-Castro
Key: 522B-4332-E8DF-86B3-8CAF

This certificate may be verified at secure.authorservices.springernature.com/certificate/verify

Nature Research Editing Service is a service from Springer Nature, one of the world's leading research, educational and professional publishers. We have been a reliable provider of high-quality editing since 2008.

Nature Research Editing Service comprises a network of more than 900 language editors with a range of academic backgrounds. All our language editors are native English speakers and must meet strict selection criteria. We require that each editor has completed or is completing a Masters, Ph.D. or M.D. qualification, is affiliated with a top US university or research institute, and has undergone substantial editing training. To ensure we can meet the needs of researchers in a broad range of fields, we continually recruit editors to represent growing and new disciplines.

Uploaded manuscripts are reviewed by an editor with a relevant academic background. Our senior editors also quality-assess each edited manuscript before it is returned to the author to ensure that our high standards are maintained.

“4) The authors mention that the used vectors contain lox sites (s. also major point 2). However, no vector map is provided to estimate what part of the vector is deleted after Cre application. Could the authors provide vector maps or at least a reference to a paper containing those maps? For the readers it would also be useful to know whether the vectors are available from Addgene.”

Addgene references were added to the Methods section in the new version of the paper, including a new reference to the paper describing the protocol (²¹, Figure 1). LoxP site positions in the vector are indicated by the green arrows in Letter Figure 7. We apologize for the inattention.

Letter Figure 7: Representative L2GOC2 (Cherry) vector backbone map (<http://www.addgene.org/27339/>)

“5) p. 23 and Fig 3G: the restriction enzyme was most likely *HaeIII* (as also said in the Table of Fig. 3G), not *HeaIII*?”

As argued by Reviewer 2, *HaeIII* was mistyped in Fig. 3G. We apologize for the inattention and modified the corresponding text.

“6) Did the authors also perform SKY analysis on the parental cells for comparison”

The earliest SKY analysis was performed on RAINBONE cells after RGB marking.

REFERENCES:

1. Lamprecht, S. *et al.* Multicolor lineage tracing reveals clonal architecture and dynamics in colon cancer. *Nature Communications* **8**, 1406 (2017).
2. Casasent, A. K. *et al.* Multiclonal Invasion in Breast Tumors Identified by Topographic Single Cell Sequencing. *Cell* **172**, 205–217.e12 (2018).
3. Bakhoun, S. F. & Landau, D. A. Cancer Evolution: No Room for Negative Selection. *Cell* **171**, 987–989 (2017).
4. Williams, M. J., Werner, B., Barnes, C. P., Graham, T. A. & Sottoriva, A. Reply: Is the evolution of tumors Darwinian or non-Darwinian? *Natl Sci Rev* **5**, 17–19 (2018).
5. Johnson, D. C. *et al.* Neutral tumor evolution in myeloma is associated with poor prognosis. *Blood* blood-2016-11-750612 (2017). doi:10.1182/blood-2016-11-750612
6. Amirouchene-Angelozzi, N., Swanton, C. & Bardelli, A. Tumor Evolution as a Therapeutic Target. *Cancer Discov* **7**, 805–817 (2017).
7. Rosner, B. *Fundamentals of Biostatistics*. (Brooks/Cole, 2010).
8. Quintana, E. *et al.* Efficient tumor formation by single human melanoma cells. *Nature* **456**, 593–598 (2008).
9. Rubio, R. *et al.* Bone environment is essential for osteosarcoma development from transformed mesenchymal stem cells. *Stem Cells* **32**, 1136–1148 (2014).
10. Patel, M. S., Shah, H. S. & Shrivastava, N. c-Myc-Dependent Cell Competition in Human Cancer Cells. *J. Cell. Biochem.* **118**, 1782–1791 (2017).
11. Giacomo, S. D. *et al.* Human Cancer Cells Signal Their Competitive Fitness Through MYC Activity. *Scientific Reports* **7**, 12568 (2017).
12. Greaves, M. & Maley, C. C. Clonal evolution in cancer. *Nature* **481**, 306–313 (2012).
13. McGranahan, N. & Swanton, C. Clonal Heterogeneity and Tumor Evolution: Past, Present, and the Future. *Cell* **168**, 613–628 (2017).

14. Waclaw, B. *et al.* A spatial model predicts that dispersal and cell turnover limit intratumour heterogeneity. *Nature* **525**, 261–264 (2015).
15. Amend, S. R., Sounak Roy, Brown, J. S. & Pienta, K. J. Ecological paradigms to understand the dynamics of metastasis. *Cancer Lett* **380**, 237–242 (2016).
16. Eichenlaub, T., Cohen, S. M. & Herranz, H. Cell Competition Drives the Formation of Metastatic Tumors in a *Drosophila* Model of Epithelial Tumor Formation. *Current Biology* **26**, 419–427 (2016).
17. Taylor, T. B., Wass, A. V., Johnson, L. J. & Dash, P. Resource competition promotes tumour expansion in experimentally evolved cancer. *BMC Evol Biol* **17**, (2017).
18. Davis, A., Gao, R. & Navin, N. Tumor evolution: Linear, branching, neutral or punctuated? *Biochim Biophys Acta* **1867**, 151–161 (2017).
19. Williams, M. J., Werner, B., Barnes, C. P., Graham, T. A. & Sottoriva, A. Identification of neutral tumor evolution across cancer types. *Nat Genet* **48**, 238–244 (2016).
20. Efremova, M. *et al.* Targeting immune checkpoints potentiates immunoediting and changes the dynamics of tumor evolution. *Nature Communications* **9**, 32 (2018).
21. Weber, K., Mock, U., Petrowitz, B., Bartsch, U. & Fehse, B. Lentiviral gene ontology (LeGO) vectors equipped with novel drug-selectable fluorescent proteins: new building blocks for cell marking and multi-gene analysis. *Gene Therapy* **17**, 511–520 (2010).

Reviewers' comments:

Reviewer #1 (Remarks to the Author):

The authors have addressed each of my prior criticisms, effectively. There remain some weaknesses intrinsic to the model of transformation, but the data can support the claims made. The use of limiting dilution-defined clones that are subsequently re-implanted certainly strengthens the arguments for their particular hypotheses. There remains the challenge that these are not natural tumors, but actually induced tumors, but the theoretical implications retain great strength.

Reviewer #2 (Remarks to the Author):

Gambera et al have addressed all major points. I have only some minor points left:

1) p. 10. L 201f: Regarding SPADE and ViSNE analyses the authors refer to Fig. 3E-F, but no SPADE or ViSNE analyses are depicted in Fig 3F.

2) Suppl. Figure 5d: The X axis is not labelled [hours].

3) The most right label in Fig 5C should obviously be $>500 \mu\text{m}$ (not $<500 \mu\text{m}$)

4) Description of Fig 5G is a bit confusing, i.e. timing does not correspond entirely with the main text. Were the cell inoculated after 60 days, and analysis was performed one week later? This referee at least got lost there...

5) Some general remark: Give the high incidence of red-green blindness, the authors might want to avoid using red and green hues wherever possible (obviously they cannot do so with their cell marking strategy, but in plots etc...)

POINT-BY-POINT RESPONSE TO REVIEWER 2

1) p. 10. L 201f: Regarding SPADE and ViSNE analyses the authors refer to Fig. 3E-F, but no SPADE or ViSNE analyses are depicted in Fig 3F.

Figure reference was modified; effectively only Fig.3E shows SPADE analysis. In the new version of the manuscript, SPADE and ViSNE analyses are referenced as Fig. 3e and Supplementary Fig. 3b-c.

2) Suppl. Figure 5d: The X axis is not labelled [hours]. Figure axis was added.

3) The most right label in Fig 5C should obviously be $>500 \mu\text{m}$ (not $<500 \mu\text{m}$). Fig. 5C label was modified.

4) Description of Fig 5G is a bit confusing, i.e. timing does not correspond entirely with the main text. Were the cell inoculated after 60 days, and analysis was performed one week later? This referee at least got lost there...

We agree with the reviewer that the description is confusing and we apologize for the mistake. Indeed, the correct timing of the experiment should be:

- Day 0: RAINBONE cells orthotopic inoculation.
- Day 0-50: In vivo imaging and disease development study.
- Day 50: Intravenous inoculation of metastatic competent clones (reseeding test).
- Day 60: In vivo imaging and mice euthanasia.

In the text, we said that mice were sacrificed one week after the intravenous administration of metastatic competent clones, but actually mice were sacrificed 10 days after. We corrected the graph and the corresponding text.

5) Some general remark: Give the high incidence of red-green blindness, the authors might want to avoid using red and green hues wherever possible (obviously they cannot do so with their cell marking strategy, but in plots etc...)

We apologize for neglecting colour blindness readers, it is certainly an important point that we have solved. As Reviewer 2 said, the RGB system is not properly compatible with daltonism. For that reason, we optimized the figures in order to make our manuscript colour-blindness friendly, and our new version of the manuscript is designed to be accessible for green-red blind readers. All data derived by confocal microscopy were not changed; most of the figures were adapted. The colour conversion was optimized manually using the plugin VischeckJ for ImageJ, which was employed to simulate colour-blind vision. Each figure was tested for deuteranopia and protanopia.

The main changes include:

- 1) We optimized colour discrimination of scale bars with a distinguishable colour code.

Example figure 1

2) We manually adapted each figure to increase colour differences/contrasts:

Example: New version of Fig.3

In detail, the principles guiding layout optimization were:

Example Fig. 2A and 2F: Red colour do not change the information presented; for that reason the colour was kept.

Example Fig. 2D: Colour-blind readers can notice the colour changes between red dominant clones and blue/yellow clones infiltrating the previous one. Notice that colour blind readers will see the change in yellow intensity. RGB system also generates an extremely variable color combination using different colour intensities. By contrast, in this specific case, green infiltrating cells cannot be visualized properly by color-blind readers and for that reason we removed the arrow indicating these cells, which were present in the previous figure version.

Fig.3 schematic: Information does not change for colour-blind readers, so no changes were made.

Fig.3E: SPADE analysis was presented in an RGB scale in the previous manuscript version. We converted the RGB (Red, Green, Blue) scale to BYR scale (Blue, Yellow, Red). This conversion preserves the information while it is equally visible for normal and colour-blind readers.

3) We were unable to convert ViSNE analysis; the conversion to the BYR scale was not effective and actually colour-blind readers can interpret the graph properly.

Example: Supplementary Fig.S3

- 4) Unfortunately, we were unable to change the colours represented in the 3D flow cytometry graphs, as FlowJo software does not allow to choose different settings. In any case, 3D cubes offer a spatial interpretation of the colours. We think that the information is preserved.

Example figure 4.

- 5) We adapted some colours to increase contrast in graphs and schematics. We also added symbols to facilitate graphs interpretation.

Example figure 5.

REVIEWERS' COMMENTS:

Reviewer #2 (Remarks to the Author):

The authors have addressed all my comments